# USP28 deletion and small-molecule inhibition destabilizes c-MYC and elicits regression of squamous cell lung carcinoma

E Josue Ruiz[1], Adan Pinto-Fernandez[2], Andrew P Turnbull[3], Linxiang Lan[1], Thomas M Charlton[2], Hannah C Scott[2], Andreas Damianou[2], George Vere[2], Eva M Riising[1], Clive Da Costa[1], Wojciech W Krajewski[3], David Guerin[4†], Jeffrey D Kearns[4‡], Stephanos Ioannidis[4§], Marie Katz[4#], Crystal McKinnon[4#], Jonathan O'Connell[4#], Natalia Moncaut[5], Ian Rosewell[5], Emma Nye[1], Neil Jones[3], Claire Heride[3], Malte Gersch[6], Min Wu[4¶], Christopher J Dinsmore[4**], Tim R Hammonds[3††], Sunkyu Kim[7], David Komander[8], Sylvie Urbe[9], Michael J Clague[9], Benedikt M Kessler[2*], Axel Behrens[1,10,11,12*]

[1]Adult stem cell laboratory, The Francis Crick Institute, London, United Kingdom; [2]Target Discovery Institute, Nuffield Department of Medicine, University of Oxford, Oxford, United Kingdom; [3]London Bioscience Innovation Centre, CRUK Therapeutic Discovery Laboratories, London, United Kingdom; [4]FORMA Therapeutics, Watertown, United Kingdom; [5]Genetic Manipulation Service, The Francis Crick Institute, London, United States; [6]Max Planck Institute of Molecular Physiology, Dortmund, Germany; [7]Incyte, Wilmington, United States; [8]Ubiquitin Signalling Division, Walter and Eliza Hall Institute of Medical Research, Royal Parade, and Department of Medical Biology, University of Melbourne, Melbourne, Australia; [9]Cellular and Molecular Physiology, Institute of Translational Medicine, University of Liverpool, Liverpool, United Kingdom; [10]Cancer Stem Cell Laboratory, Institute of Cancer Research, London, United Kingdom; [11]Imperial College, Division of Cancer, Department of Surgery and Cancer, London, United Kingdom; [12]Convergence Science Centre, Imperial College, London, United Kingdom

*For correspondence: benedikt.kessler@ndm.ox.ac.uk (BMK); axel.behrens@icr.ac.uk (AB)

Present address: †Constellation Pharmaceuticals, Cambridge, United States; ‡Novartis Institutes for BioMedical Research, Cambridge, United States; §IFM Therapeutics, MA 02116, United States; #Valo Health, MA 02116, United States; ¶Disc Medicine, Cambridge, United States; **Kronos Bio, Inc, Cambridge, United States; ††Locki Therapeutics, London Bioscience Innovation Centre, London, United Kingdom

**Abstract** Lung squamous cell carcinoma (LSCC) is a considerable global health burden, with an incidence of over 600,000 cases per year. Treatment options are limited, and patient's 5-year survival rate is less than 5 %. The ubiquitin-specific protease 28 (USP28) has been implicated in tumourigenesis through its stabilization of the oncoproteins c-MYC, c-JUN, and Δp63. Here, we show that genetic inactivation of *Usp28*-induced regression of established murine LSCC lung tumours. We developed a small molecule that inhibits USP28 activity in the low nanomole range. While displaying cross-reactivity against the closest homologue USP25, this inhibitor showed a high degree of selectivity over other deubiquitinases. USP28 inhibitor treatment resulted in a dramatic decrease in c-MYC, c-JUN, and Δp63 proteins levels and consequently induced substantial regression of autochthonous murine LSCC tumours and human LSCC xenografts, thereby phenocopying the effect observed by genetic deletion. Thus, USP28 may represent a promising therapeutic target for the treatment of squamous cell lung carcinoma.

## Introduction

Lung cancer is the leading cause of cancer death worldwide. Based on histological criteria, lung cancer can be subdivided into non-small cell lung cancer (NSCLC) and the rarer small cell lung cancer. The most common NSCLCs are lung adenocarcinoma (LADC) and lung squamous cell carcinoma (LSCC), with large cell carcinoma being less commonly observed. Progress has been made in the targeted treatment of LADC, largely due to the development of small-molecule inhibitors against EGFR, ALK, and ROS1 (*Cardarella and Johnson, 2013*). However, no targeted treatment options exist for LSCC patients (*Hirsch et al., 2017*; *Novello et al., 2014*). Consequently, despite having limited efficacy on LSCC patient survival, platinum-based chemotherapy remains the cornerstone of current LSCC treatment (*Fennell et al., 2016*; *Isaka et al., 2017*; *Scagliotti et al., 2008*). Therefore, there is an urgent need to identify novel druggable targets for LSCC treatment and to develop novel therapeutics.

The *Fbxw7* protein product F-box/WD repeat-containing protein 7 (FBW7) is the substrate recognition component of an SCF-type ubiquitin ligase, which targets several well-known oncoproteins, including c-MYC, NOTCH, and c-JUN, for degradation (*Davis et al., 2014*). These oncoproteins accumulate in the absence of FBW7 function, and genetic analyses of human LSCC samples revealed common genomic alterations in *Fbxw7* (*Cancer Genome Atlas Research Network, 2012*; *Kan et al., 2010*). In addition, FBW7 protein is undetectable by immunohistochemistry (IHC) in 69 % of LSCC patient tumour samples (*Ruiz et al., 2019*). Genetically engineered mice (GEM) harbouring loss of *Fbxw7* concomitant with *Kras^{G12D}* activation (KF mice) develop LSCC with 100 % penetrance and short latency, as well as LADC (*Ruiz et al., 2019*). Thus, FBW7 is an important tumour suppressor in both human and murine lung cancer.

The deubiquitinase USP28 opposes FBW7-mediated ubiquitination of the oncoproteins c-MYC and c-JUN, thereby stabilizing these proteins (*Popov et al., 2007*). In a murine model of colorectal cancer, deleting *Usp28* reduced size of established tumours and increased lifespan (*Diefenbacher et al., 2014*). Therefore, targeting USP28 in order to destabilize its substrates represents an attractive strategy to inhibit the function of c-MYC and other oncogenic transcription factors that are not amenable to conventional inhibition by small molecules.

Here, we describe the characterization of a novel USP28 inhibitory compound (USP28i) and the genetic as well as chemical validation of USP28 as a promising therapeutic target for LSCC tumours. Using an Frt-Flp and Cre-LoxP dual recombinase system (*Schönhuber et al., 2014*), we show that *Usp28* inactivation in established LSCC results in dramatic tumour regression. Importantly, USP28i treatment recapitulates LSCC regression in both mouse models and human LSCC xenografts. Absence or inhibition of USP28 resulted in a dramatic decrease in the protein levels of c-MYC, c-JUN, and Δp63, providing a potential mechanism of action for USP28i. Therefore, USP28 inhibition should be a strong candidate for clinical evaluation, particularly given the paucity of currently available therapy options for LSCC patients.

## Results

### USP28 is required to maintain protein levels of c-MYC, c-JUN, and Δp63 in LSCC

To gain insights into the molecular differences between LADC and LSCC, we investigated the expression of MYC in these common NSCLCs subtypes. MYC was transcriptionally upregulated in human LSCC compared to healthy lung tissue or LADC tumours (*Figure 1A*). Quantitative polymerase chain reaction (qPCR) analysis on an independent set of primary human lung biopsy samples confirmed that MYC is highly expressed in LSCC tumours compared with normal lung tissue (*Figure 1B*). Moreover, IHC staining on primary lung tumours confirmed a significant abundance of c-MYC protein in LSCC samples (*Figure 1C and D*). Also, Δp63 and c-JUN, critical factors in squamous cell identity and tumour maintenance, respectively, showed higher protein levels in LSCC compared to LADC tumours (*Figure 1C and D*). Individual downregulation of c-MYC, c-JUN, and Δp63 by small interfering RNA (siRNA) resulted in a significant reduction of cell growth in four independent human LSCC cell lines (*Figure 1E*, *Figure 1—figure supplement 1A-C*).

As c-MYC, c-JUN, and Δp63 protein levels are controlled by the deubiquitinase USP28 (*Popov et al., 2007*; *Prieto-Garcia et al., 2020*), we analysed its expression in publicly available datasets (The Cancer Genome Atlas). We observed that 25 % of human LSCC cases show gain-of-function

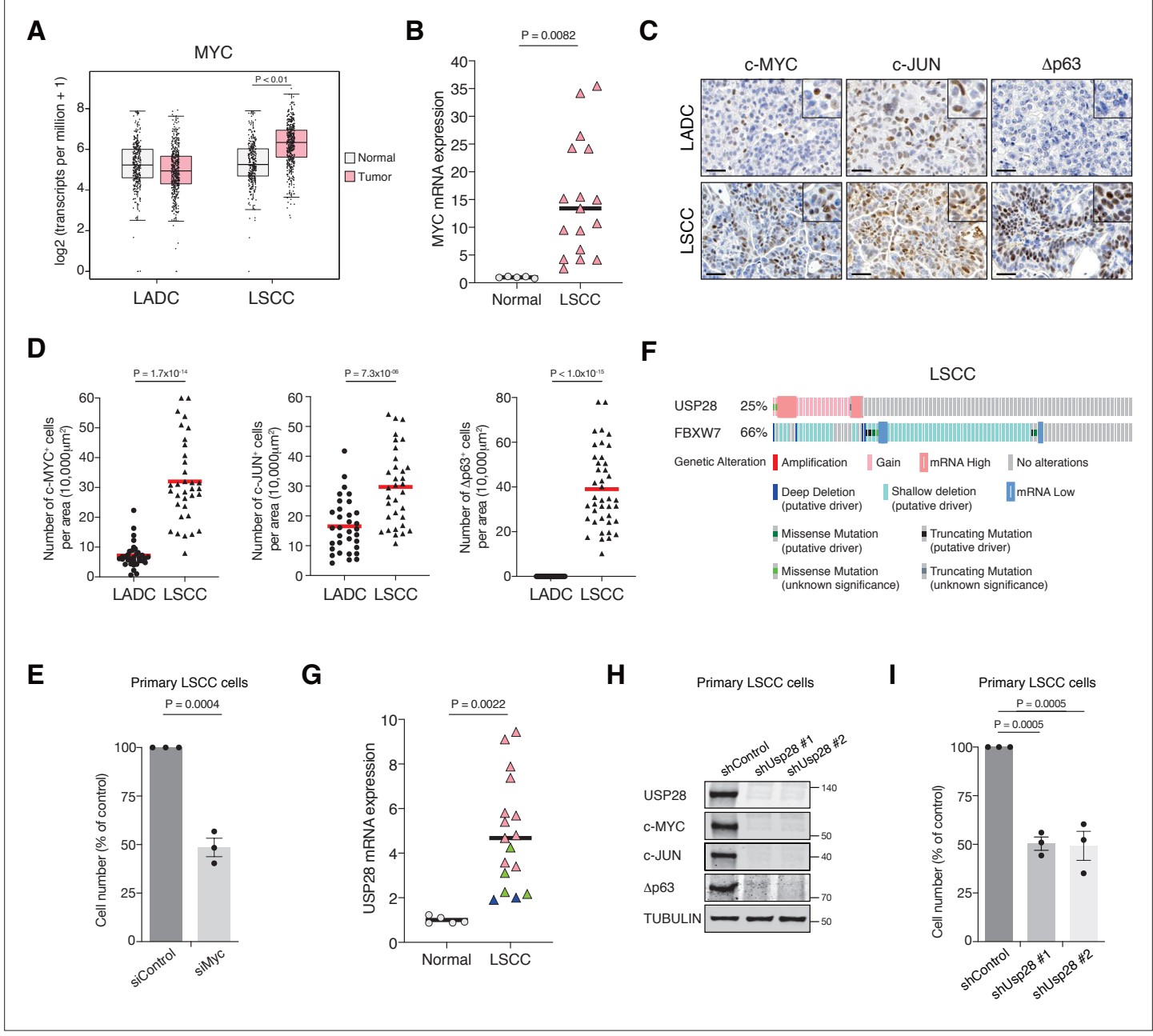

**Figure 1.** MYC, JUN, and Δp63 are highly expressed in lung squamous cell carcinoma (LSCC) tumours. (**A**) Expression of *MYC* in human lung adenocarcinoma (LADC, n = 483), lung squamous cell carcinoma (LSCC, n = 486), and normal non-transformed tissue (normal LSCC = 338, normal LADC = 347). In box plots, the centre line reflects the median. Data from TCGA and GTEx were analysed using GEPIA software. (**B**) Relative mRNA expression of *MYC* in normal lung tissue (n = 5) and LSCC (n = 17) patient samples from the Cordoba Biobank measured by RT-PCR. The p value was calculated using the Student's two-tailed t test. Plots indicate mean. (**C**) Representative LADC and LSCC tumours stained with c-MYC, c-JUN, and Δp63 antibodies. Scale bars, 30 µm. (**D**) Quantification of c-MYC+ (LADC n = 33, LSCC n = 34), c-JUN+ (LADC n = 33, LSCC n = 33), and Δp63+ cells (LADC n = 41, LSCC n = 41) in LADC and LSCC tumours. Plots indicate mean. Student's two-tailed t test was used to calculate p values. (**E**) Graph showing the difference in cell proliferation between control and MYC-depleted KF LSCC cells (n = 3). Graph indicates mean ± SEM. Student's two-tailed t test was used to calculate p values. (**F**) Genetic alterations in ubiquitin-specific protease 28 (*USP28*) and *FBXW7* genes in human LSCC. Each column represents a tumour sample (n = 178). Data from TCGA were analysed using cBioportal software. (**G**) Relative mRNA expression of *USP28* in normal lung tissue (n = 5) and LSCC (n = 17) patient samples from the Cordoba Biobank measured by RT-PCR. The p value was calculated using the Student's two-tailed t test. Plots indicate mean. See also *Figure 1—figure supplement 1B*. (**H**) shRNA-mediated knockdown of *Usp28* decreases c-MYC, c-JUN, and Δp63 protein levels in primary KF LSCC cells. (**I**) Graph showing the difference in cell proliferation between control and *Usp28*-depleted KF LSCC cells (n = 3). Graph indicates mean ± SEM. One-way analysis of variance (ANOVA) with Dunnett's multiple comparisons test was used to calculate p values. Source data for B, D, E, G,

*Figure 1 continued on next page*

*Figure 1 continued*

and I.

The online version of this article includes the following source data and figure supplement(s) for figure 1:

**Source data 1.** c-MYC, c-JUN, Dp63 and USP28 are highly expressed in LSCC tumours.

**Figure supplement 1** c-MYC, c-JUN and dp63 knockdown affects LSCC cell line growth.

**Figure supplement 1—source data 1.** c-MYC, c-JUN and Dp63 knockdown affect proliferation of human LSCC.

**Figure supplement 2.** USP28 expression in LSCC tumours.

**Figure supplement 2—source data 1.** USP28 copy-number vs mRNA expression in human LSCC patients.

alterations in *USP28* (*Figure 1F*). In addition, a positive correlation between *USP28* copy-number and mRNA expression was found in the same datasets (*Figure 1—figure supplement 2A*). Interestingly, qPCR and IHC analysis on human LSCC samples revealed that low *USP28* mRNA levels correlated with low USP28 protein levels and likewise, high/moderate mRNA levels also correlated with high USP28 protein levels (*Figure 1G*, *Figure 1—figure supplement 2B*). Since USP28 is involved in Δp63, c-JUN, and c-MYC stabilization and higher expression of USP28 is associated with a significantly shorter survival time (*Prieto-Garcia et al., 2020*), we targeted its expression. *Usp28* downregulation by shRNA resulted in a significant reduction in c-MYC, c-JUN, and Δp63 protein levels in LSCC primary tumour cells and reduced LSCC cell growth (*Figure 1H and I*). Thus, targeting USP28 in order to destabilize its substrates represents a rational strategy to target tumour cells that rely on oncogenic transcription factors that are currently not druggable by small molecules.

## Generation of a pre-clinical dual recombinase lung cancer mouse model

Recently, *Usp28* was shown to be required for the initiation of lung tumours in the *Rosa26-Cas9* sgRNA *Kras^{G12D}*; *Trp53*; *Lkb1* model (*Prieto-Garcia et al., 2020*). However, a meaningful pre-clinical model requires targeting the therapeutic candidate gene in existing growing lung tumours. Thus, to assess the function of *Usp28* in established tumours, we developed a new GEM model to temporally and spatially separate tumour development from target deletion by using two independent recombinases: FLP and CreERT. In this model, LSCC and LADC formation is initiated by *Kras^{G12D}* activation and *Fbxw7* deletion using FLP recombinase, and the Cre-loxP system can then be used for inactivation of *Usp28^{flox/flox}* in established tumours. To allow conditional FRT/FLP-mediated inactivation of *Fbxw7* function, we inserted two FRT sites flanking exon 5 of the endogenous *Fbxw7* gene in mice to generate a *Fbxw7^{FRT/FRT}* allele that can be deleted by FLP recombinase (*Figure 2—figure supplement 1A and B*). Expression of FLP recombinase resulted in the deletion of *Fbxw7* exon 5, which could be detected by PCR (*Figure 2—figure supplement 1B*). The resulting strain, *Fbxw7^{FRT/FRT}*, was crossed to *FRT-STOP-FRT (FSF)-Kras^{G12D}* mice to generate *Kras^{FSF-G12D}*; *Fbxw7^{FRT/FRT}* (KF-Flp model).

## USP28 is an effective therapeutic target for LSCC, but not KRas^{G12D}; Trp53 mutant LADC tumours

The KF-Flp strain described above was crossed with *Rosa26^{FSF-CreERT}*; *Usp28^{flox/flox}* mice to generate the KFCU model (*Figure 2A*). KFCU tumour development was monitored by computed tomography (CT) scans. At 10–11 weeks post-infection with FLP recombinase-expressing recombinant adenoviruses, animals displayed lesions in their lungs. At this time point, we confirmed by histology that KFCU mice develop both LADC and LSCC tumours (*Figure 2—figure supplement 1C*). As expected (*Ruiz et al., 2019*), KFCU LADC lesions occurred in alveolar tissue and were positive for SFTPC and TTF1. KFCU LSCC tumours occurred mainly in bronchi (rarely manifesting in the alveolar compartment) and expressed CK5 and Δp63. Next, animals displaying lung tumours were exposed to tamoxifen to activate the CreERT protein and delete the conditional *Usp28* floxed alleles (*Figure 2A*, *Figure 2—figure supplement 1D*). Mice transiently lost body weight during the initial tamoxifen treatment but recovered a few days later (*Figure 2—figure supplement 1E*). Although the loss of USP28 expression decreased LADC tumour size, it did not reduce the number of LADC tumours (*Figure 2B–D*). In contrast, histological examination of KFCU mice revealed a clear reduction in the numbers of LSCC lesions in *Usp28*-deleted lungs (*Figure 2F*, *Figure 2—figure supplement 1D*). As well as a significant reduction in tumour number, the few CK5-positive LSCC lesions that remained were substantially

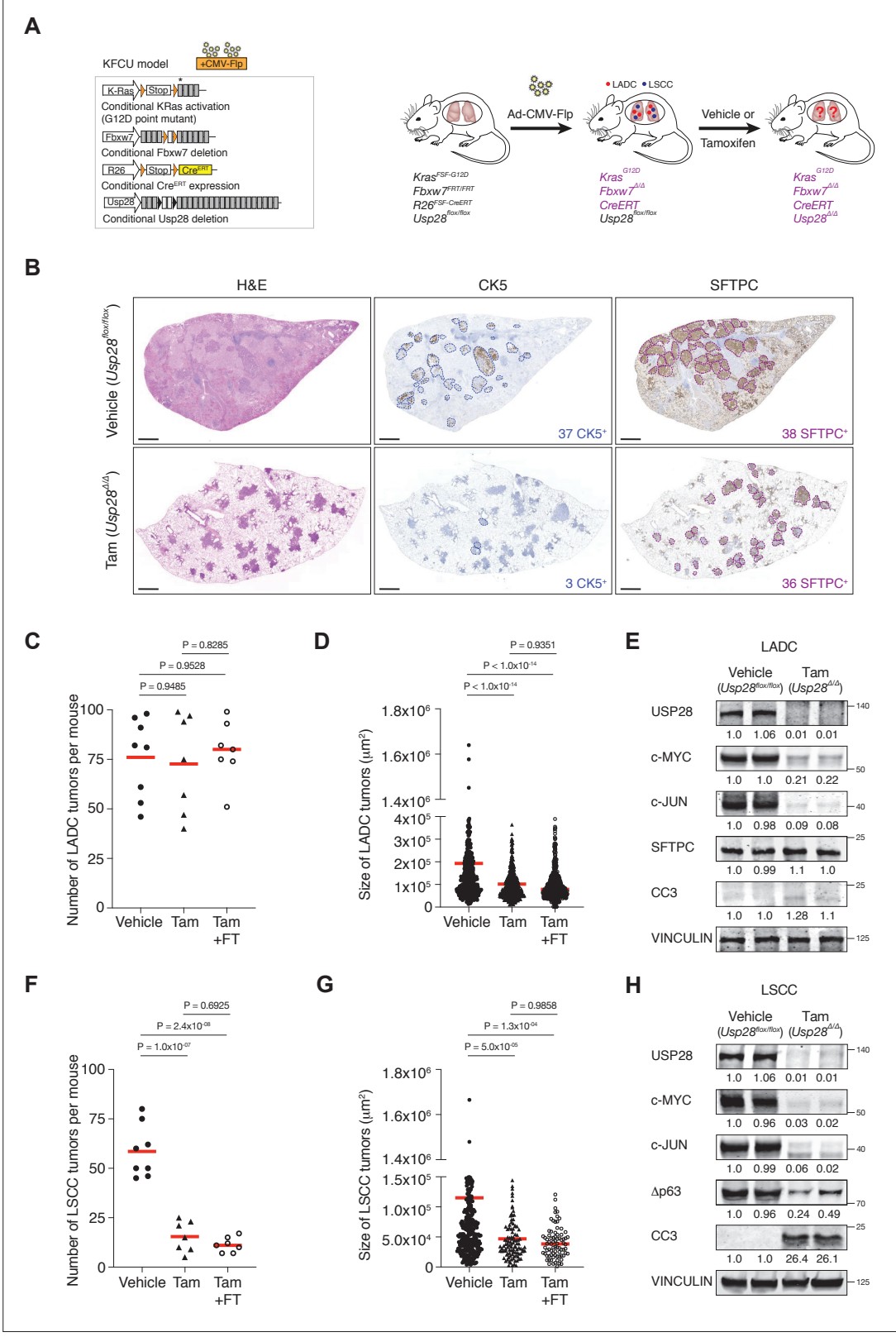

**Figure 2.** Ubiquitin-specific protease 28 (USP28) is an effective therapeutic target for lung squamous cell carcinoma (LSCC) tumours. (**A**) Schematic representation of the KFCU (<u>K</u>ras<sup>FSF-G12D</sup>; <u>F</u>bxw7<sup>FRT/FRT</sup>; Rosa26<sup>FSF-C</sup><sup>reERT</sup>; <u>U</u>sp28<sup>flox/flox</sup>) model and experimental approach used to deplete conditional *Usp28* alleles in established lung tumours. (**B**) Lung histology of animals treated as in A, showing both LSCC (CK5⁺) and lung adenocarcinoma

*Figure 2 continued*

(LADC) (SFTPC⁺) tumours in mice receiving vehicle but few LSCC lesions in mice receiving tamoxifen. Scale bars, 1000 μm. (**C**) Quantification of LADC tumours in vehicle-, tamoxifen-, and tamoxifen+ FT206-treated KFCU mice. Plots indicate mean. One-way analysis of variance (ANOVA) with Tukey's multiple comparisons test was used to calculate p values (n = 8 vehicle, n = 7 tamoxifen, n = 7 tamoxifen+ FT206). (**D**) Quantification of LADC tumour size in vehicle-, tamoxifen-, and tamoxifen+ FT206-treated KFCU mice. Plots indicate mean. One-way ANOVA with Tukey's multiple comparisons test was used to calculate p values (n = 466 vehicle, n = 434 tamoxifen, n = 503 tamoxifen+ FT206). (**E**) Immunoblot analysis of LADC tumours probed for USP28, c-MYC, c-JUN, SFTPC, cleaved caspase-3 (CC3). VINCULIN is shown as loading control. (**F**) Quantification of LSCC tumours in vehicle-, tamoxifen-, and tamoxifen+ FT206-treated KFCU mice. Plots indicate mean. One-way ANOVA with Tukey's multiple comparisons test was used to calculate p values (n = 8 vehicle, n = 7 tamoxifen, n = 7 tamoxifen+ FT206). (**G**) Quantification of LSCC tumour size in vehicle-, tamoxifen-, and tamoxifen+ FT206-treated KFCU mice. Plots indicate mean. One-way ANOVA with Tukey's multiple comparisons test was used to calculate p values (n = 326 vehicle, n = 103 tamoxifen, n = 79 tamoxifen+ FT206). (**H**) *Usp28* deletion induces apoptotic cell death (CC3) and decreases c-MYC, c-JUN, and Δp63 protein levels in LSCC lesions. Source data for C, D, F, and G.

The online version of this article includes the following source data and figure supplement(s) for figure 2:

**Source data 1.** Quantification of LADC and LSCC tumours in the KFCU model.

**Figure supplement 1.** Gene 1 targeting strategy to generate a Fbxw7 FRT/FRT allele that can be deleted by Flp recombinase.

**Figure supplement 1—source data 1.** Body weights of animals treated with Vehicle, Tamoxifen (Tam) or Tamoxifen+FT206.

smaller than control tumours (*Figure 2G*). Measurement of the size of 429 individual KFCU LSCC tumours (326 vehicle-treated and 103 tamoxifen-treated) showed an average size of $11.4 \times 10^4$ μm$^2$ in the vehicle arm versus $4.6 \times 10^4$ μm$^2$ in the tamoxifen arm (*Figure 2G*). Thus, *Usp28* inactivation significantly reduces both the number and the size of LSCC tumours.

To get insights into LSCC tumour regression, we focused on USP28 substrates. Immunoblotting analysis revealed that *Usp28* deletion resulted in apoptotic cell death (cleaved caspase-3; CC3). Δp63 protein levels were reduced, but c-JUN and c-MYC protein became undetectable (*Figure 2H*, *Figure 2—figure supplement 1F*). *Usp28* deletion also decreased c-JUN and c-MYC levels in KFCU LADC lesions, although the reduction in c-MYC protein levels was significantly less pronounced than observed in LSCC (*Figure 2E*). Strikingly, elimination of *Usp28* has little effect, if any, on apoptotic cell death, as determined by its inability to induce CC3 in LADC lesions. Thus, these data suggest that USP28 and its substrates are required for the maintenance of LSCC tumours.

To further investigate the role of USP28 in LADC, we studied the consequences of *Usp28* deletion in a second LADC genetic model. We used Flp-inducible oncogenic *Kras* activation combined with *Trp53* deletion (*Kras^FSF-G12D^* and *Trp53^FRT/FRT^* or KP-Flp model) (*Schönhuber et al., 2014*). The KP-Flp mice were crossed to a conditional *Usp28^flox/flox^* strain together with an inducible CreERT recombinase knocked in at the *Rosa26* locus and an mT/mG reporter allele (KPCU mice; *Figure 3A*). After intratracheal adeno-CMV-Flp virus instillation, *Usp28* was deleted in KPCU animals displaying lung tumours by CT (*Figure 3A*). Loss of USP28 expression in this second LADC model also did not result in a reduction of LADC tumour number and size (*Figure 3C and D*). Successful CreERT recombination was verified using lineage tracing (GFP staining) and deletion of *Usp28^flox/flox^* alleles was further confirmed by BaseScope assays (*Figure 3B and E*). Therefore, also these data argue against an important role for USP28 in LADC tumours.

## Generation of a new USP28 inhibitor: selectivity and cellular target engagement

The finding that USP28 plays a key role in LSCC tumour maintenance prompted us to identify small-molecule inhibitors against this deubiquitinase. A small-molecule discovery campaign based on the ubiquitin-rhodamine cleavable assay (*Turnbull et al., 2017*) yielded a panel of compounds sharing a thienopyridine carboxamide chemical scaffold with inhibitory selectivity for USP28 and USP25 (*Guerin et al., 2017*; *Guerin et al., 2020*; *Zablocki et al., 2019*). The compound FT206 (*Figure 4A*) represents a different chemical class from the benzylic amino ethanol-based inhibitors described previously (*Wrigley et al., 2017*). Quantitative structure-activity relationship was used to develop compound

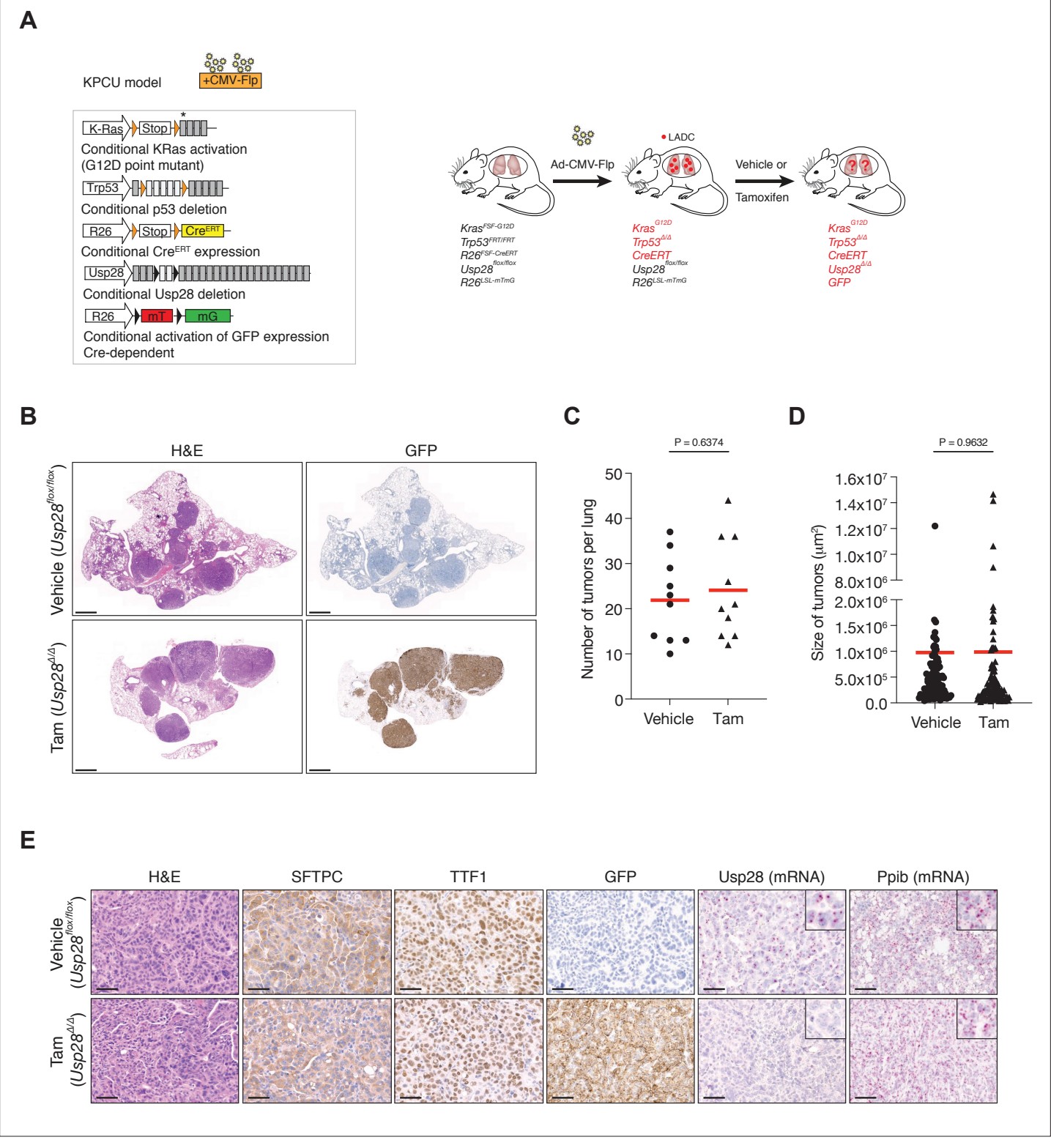

**Figure 3.** Ubiquitin-specific protease 28 (USP28) is not a therapeutic target for advanced KRas[G12D]; Trp53 mutant tumours. (**A**) Schematic representation of the KPCU (KRas[FSF-G12D]; Trp53[FRT/FRT]; Rosa26[FSF-C[reERT]]; Usp28[flox/flox]; Rosa26[LSL-mTmG]) model and experimental approach used. At 10 weeks post-infection, KPCU mice were treated with vehicle or tamoxifen. (**B**) Representative images of H&E (left) and GFP (right) stains from mice of the indicated treatments. Scale bar, 1000 μm. (**C**) Quantification of mouse lung adenocarcinoma (LADC) tumours in the KPCU model. Plots indicate mean. Student's two-tailed t test was used to calculate p values (n = 10 vehicle, n = 10 tamoxifen). (**D**) Quantification of LADC tumour size in vehicle- and tamoxifen-treated KPCU

*Figure 3 continued on next page*

Figure 3 continued

mice. Plots indicate mean. Student's two-tailed t test was used to calculate p values (n = 110 vehicle, n = 130 tamoxifen). (**E**) Representative images illustrating histological analysis of lung lesions in KPCU mice, treated with vehicle or tamoxifen. H&E, SFTPC, TTF1, GFP immunohistochemistry staining and in situ hybridization of *USP28* and *PPIB* mRNA expression. Scale bars, 50 μm. Source data for C and D.

The online version of this article includes the following source data for figure 3:

**Source data 1.** Quantification of LADC tumours in the KPCU model.

derivative FT206 that was most optimal in terms of drug metabolism and pharmacokinetic properties while preserving potency and selectivity towards USP28/25 (*Zablocki et al., 2019*). To confirm FT206 cellular target engagement, we used a Ub activity-based probe (ABP) assay (*Altun et al., 2011*; *Clancy et al., 2021*; *Panyain et al., 2020*; *Turnbull et al., 2017*). ABPs can assess DUB enzyme activity in a cellular context. DUB inhibition leads to displacement of the ABP, resulting in a molecular weight shift measurable by SDS-PAGE and immunoblotting against USP28/25. Using this approach, we found that the compound FT206 interferes with USP28/25 probe labelling (USP-ABP versus USP) in LSCC H520 cell extracts ($EC_{50}$ ~300–1000 nM, *Figure 4B*) and intact cells ($EC_{50}$ ~1–3 μM, *Figure 4C*). In contrast to FT206, AZ1, a different USP28 inhibitor (*Wrigley et al., 2017*), based on a benzylic amino ethanol scaffold, appeared to exert lower potency towards USP28 ($EC_{50}$ >30 μM) and selectivity for USP25 ($EC_{50}$ ~10–30 μM) (*Figure 4—figure supplement 1A*). To address compound selectivity more widely, we combined the ABP assay with quantitative mass spectrometry (activity-based probe profiling [ABPP]) to allow the analysis of the cellular active DUBome (*Benns et al., 2021*; *Jones et al., 2021*; *Pinto-Fernández et al., 2019*). When performing such assay in human LSCC cells, we were able to profile 28 endogenous DUBs, revealing a remarkable USP28/25 selectivity for FT206 in a dose-dependent manner (*Figure 4D*).

To further evaluate the efficacy of FT206 in targeting USP28, we tested its ability to modulate the ubiquitination status of endogenous USP28 substrates. The ubiquitination levels of c-MYC and c-JUN increased upon FT206 and MG132 co-treatment (*Figure 4—figure supplement 1B*), confirming that FT206 blocks USP28-mediated deubiquitination of its substrates. The ubiquitination level of USP28 also increased upon FT206 treatment (*Figure 4—figure supplement 1B*), which is consistent with previous observations where the enzymatic activity of DUBs can function to enhance their own stability (*de Bie and Ciechanover, 2011*). Consequently, treatment of LSCC tumour cells with FT206 resulted in reduced c-MYC, c-JUN, Δp63, and USP28 protein levels, which were restored upon addition of MG132 (*Figure 4E*, *Figure 4—figure supplement 1C*).

Finally, FT206 treatment impaired LSCC cell growth (*Figure 4F*). However, in a USP28-depleted background, FT206 neither affected cell growth nor reduced c-MYC protein levels (*Figure 4—figure supplement 1D*). Thus, this data suggests that the effects of FT206 are mediated by USP28.

## Pharmacological inhibition of USP28 is well tolerated in mice and induced LSCC tumour regression

We next evaluated the therapeutic potential of the USP28 inhibitor FT206 using the $Kras^{LSL-G12D}$; $Fbxw-7^{flox/flox}$ model (KF mice), which develop both LADC and LSCC tumour types (*Ruiz et al., 2019*). Nine weeks after adeno-CMV-Cre virus infection, when mice had developed lung tumours, we started treatment with USP28 inhibitor at 75 mg/kg, three times a week for 5 weeks (*Figure 5A*). FT206 administration had no noticeable adverse effects and treated mice maintained normal body weight (*Figure 5—figure supplement 1A and B*). Consistent with the effects observed by genetic *Usp28* inactivation (*Figure 2C*), the number of KF LADC lesions was not affected by USP28 inhibition via FT206 treatment (*Figure 5B–D*). By contrast, we found that FT206 effectively reduced LSCC tumour number by 68 % (31–10 LSCC tumours, *Figure 5B and E*). Moreover, measurement of 252 individual KF LSCC mutant tumours (156 vehicle-treated and 96 FT206-treated lesions) showed a significant reduction of over 45 % in tumour size upon FT206 treatment: an average of $8.5 \times 10^4$ μm² in the vehicle arm versus $4.5 \times 10^4$ μm² in the FT206 cohort (*Figure 5F*). Thus, USP28 inhibition by FT206 leads to a dramatic reduction in the numbers of advanced LSCC tumours, and the small number of remaining LSCC lesions is significantly reduced in size, resulting in a reduction of total LSCC burden of over 85 % by single agent treatment.

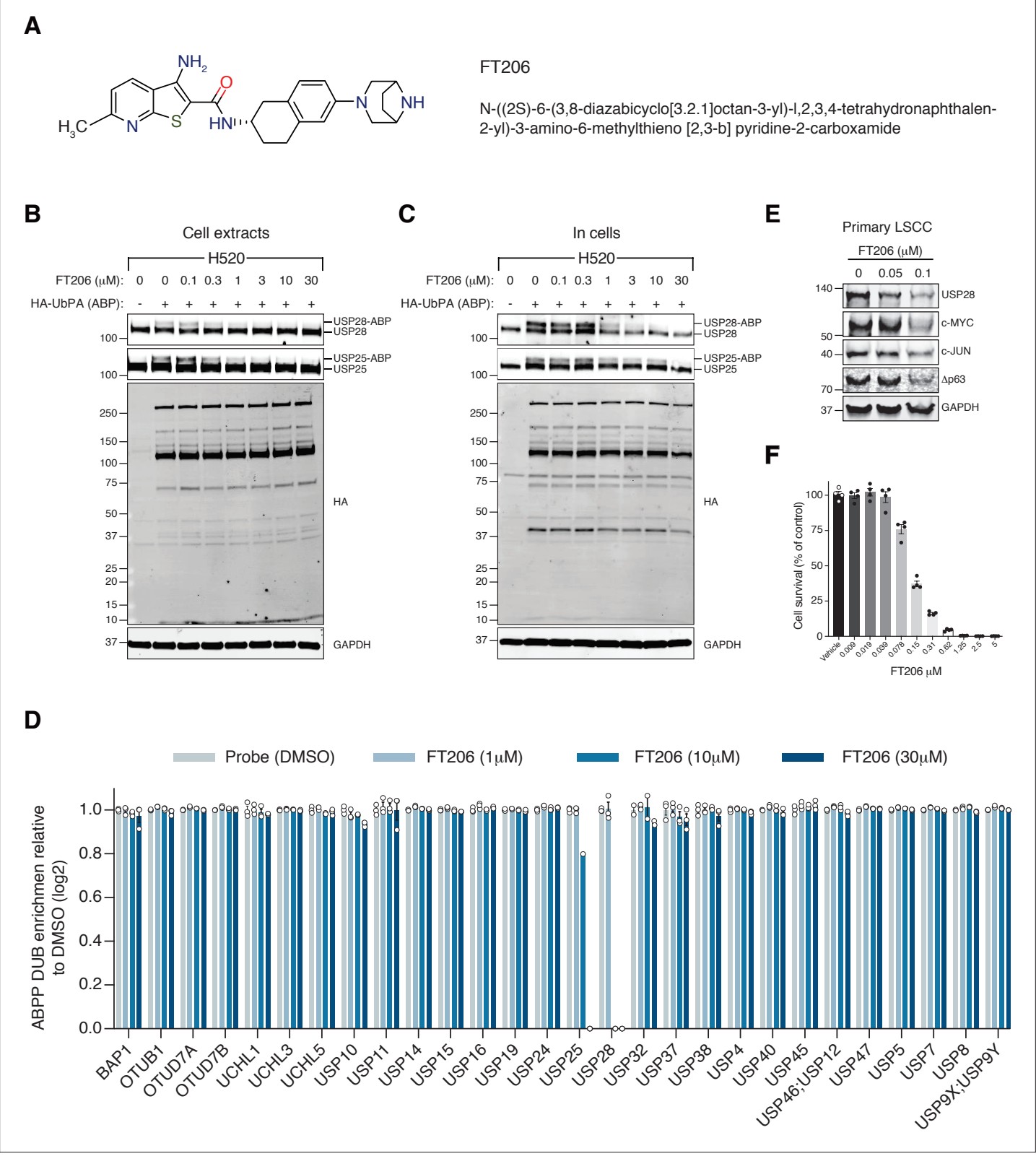

**Figure 4.** Ubiquitin-specific protease 28 (USP28) inhibitor selectivity and cellular target engagement. (**A**) Structure of small-molecular inhibitor FT3951206/CRT0511973 (FT206). (**B**) Cellular DUB profiling in NCI-H520 lung squamous cell carcinoma (LSCC) cell extracts incubated with the indicated concentrations of FT206 prior to labelling with HA-UbPA, SDS-PAGE, and analysis by Western blotting. Inhibitor potency was reflected by competition with USP28/25-ABP (activity-based probe) adduct formation. (**C**) Cellular DUB profiling in NCI-H520 LSCC cells incubated with the indicated

*Figure 4 continued on next page*

*Figure 4 continued*

concentrations of FT206, lysed extracts labelled with HA-UbPA, and analysed as in B. (**D**) Activity-based probe profiling (ABPP) demonstrating the cellular DUB selectivity profile of cpd FT206 by quantitative mass spectrometry analysis at different inhibitor concentrations. Graph indicates mean ± SEM. (**E**) USP28 inhibition using FT206 (50 and 100 nM) reduces c-MYC, c-JUN, and Δp63 protein levels in primary KF LSCC cells. (**F**) USP28 inhibition using FT206 decreases cell proliferation in KF LSCC cells (n = 4). Graph indicates mean ± SEM. Source data for F.

The online version of this article includes the following source data and figure supplement(s) for figure 4:

**Source data 1.** Activity-based Probe Profiling (ABPP) showing the cellular DUB selectivity profile of FT206 assessed by quantitative mass spectrometry.

**Source data 2.** FT206 decreases cell proliferation in LSCC cells.

**Figure supplement 1.** USP28 inhibitor targets USP28/25 and ubiquitylation levels of c-MYC, c-JUN and USP28.

**Figure supplement 1—source data 1.** Cell proliferation in control, FT206-treated and USP28-depleted LSCC cells.

In line with the effects found by genetic *Usp28* deletion, treatment of KF mice with FT206 also resulted in reduced Δp63, c-JUN, and c-MYC protein levels (*Figure 5G*). Consequently, FT206 treatment led to a substantial increase in the number of CC3-positive cells in LSCC while LADC cells were not significantly affected, indicating that USP28 inhibition causes apoptotic cell death of LSCC tumour cells (*Figure 5H and I*).

Finally, to further confirm the specificity of FT206, KFCU mice pre-exposed to tamoxifen to delete the conditional *Usp28* floxed alleles were further treated with the USP28 inhibitor FT206. In this setting, USP28 inhibition did not result in either a further reduction of LADC and LSCC lesions or body weight loss (*Figure 2C and F*, *Figure 2—figure supplement 1E*), suggesting that FT206 targets specifically USP28.

## USP28 inhibition causes dramatic regression of human LSCC xenograft tumours

To determine whether the promise of USP28 as a target in mouse lung cancer models can be translated to a human scenario, we established human xenograft tumour models. siRNA-mediated *USP28* but not *USP25* depletion, and USP28 inhibitor treatment, considerably reduced protein levels of Δp63, c-JUN, and c-MYC and impaired growth in human LSCC tumour cells (*Figure 6A–C*, *Figure 6—figure supplement 1A and B*). In contrast, FT206 treatment had marginal effects on c-MYC and c-JUN protein levels in human LADC cells and in USP28 mutant LSCC cells (*Figure 6—figure supplement 1C–1E*). Crucially, FT206 led to a remarkable growth impairment of xenografts derived from three independent human LSCC cell lines (*Figure 6D–I*), which was accompanied with a strong reduction of c-MYC protein levels (*Figure 6J–L*). In summary, these data suggest that USP28 pharmacological intervention is a promising therapeutic option for human LSCC patients.

## Discussion

Unlike for LADC, there are few approved targeted therapies against LSCC. Consequently, despite its limited effectiveness on disease progression and prognosis, patients with LSCC receive the same conventional platinum-based chemotherapy today as they would have received two decades ago (*Fennell et al., 2016*; *Gandara et al., 2015*; *Isaka et al., 2017*; *Liao et al., 2012*; *Scagliotti et al., 2008*). c-MYC is a transcription factor that orchestrates a potent pro-cancer programme across multiple cellular pathways. As c-MYC is often overexpressed in late-stage cancer, targeting it for degradation is an attractive strategy in many settings. The term 'undruggable' was coined to describe proteins that could not be targeted pharmacologically. Many desirable targets in cancer fall into this category, including the c-MYC oncoprotein, and pharmacologically targeting these intractable proteins is a key challenge in cancer research.

The deubiquitylase family of enzymes have emerged as attractive drug targets, which can offer a means to destabilize client proteins that might otherwise be undruggable (*Schauer et al., 2020*). The deubiquitinase USP28 was known to remove FBW7-mediated ubiquitination of, and thereby stabilize, the oncoprotein c-MYC (*Popov et al., 2007*). Importantly, mice lacking *Usp28* are healthy (*Knobel et al., 2014*), suggesting that USP28 is dispensable for normal physiology and homeostasis.

In the current study, we identified a requirement for USP28 for the maintenance of murine and human LSCC tumours. In agreement with the absence of major phenotypes in the *Usp28* knockout

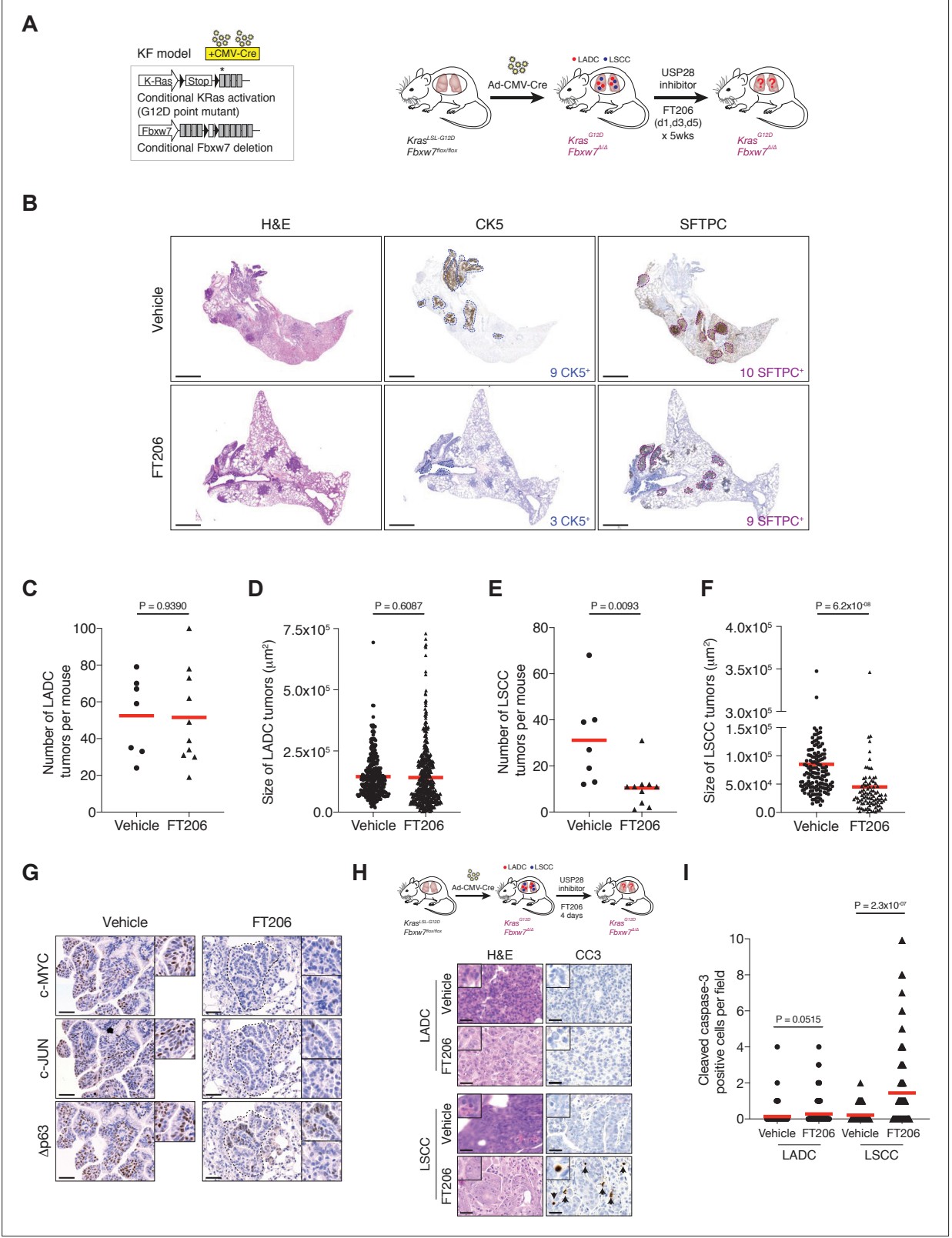

**Figure 5.** Pharmacological ubiquitin-specific protease 28 (USP28) inhibition reduces c-MYC, c-JUN, and Δp63 protein levels in mouse lung squamous cell carcinoma (LSCC) tumours and induces tumour cell death. (**A**) Scheme depicting experimental design for in vivo test of FT206 (75 mg/kg), three times a week for 5 weeks. (**B**) Lung histology of animals treated as in A, showing both LSCC (CK5⁺) and lung adenocarcinoma (LADC) (SFTPC⁺) tumours in *KRas^LSL-G12D*; *Fbxw7^f/f* (KF) mice receiving vehicle but few LSCC lesions in mice receiving FT206. Scale bars, 1000 μm. (**C**) Quantification of LADC

*Figure 5 continued on next page*

*Figure 5 continued*

tumours per animal in vehicle- and FT206-treated KF mice. Plots indicate mean. p Values calculated using Student's two-tailed t test (n = 7 vehicle, n = 10 FT206). (**D**) Quantification of LADC tumour size in vehicle- and FT206-treated KF mice. Plots indicate mean. Student's two-tailed t test was used to calculate p values (n = 304 vehicle, n = 481 FT206). (**E**) Quantification of LSCC tumours per animal in vehicle- and FT206-treated KF mice. Plots indicate mean. p Values calculated using Student's two-tailed t test (n = 7 vehicle, n = 10 FT206). (**F**) Quantification of LSCC tumour size in vehicle- and FT206-treated KF mice. Plots indicate mean. Student's two-tailed t test was used to calculate p values (n = 156 vehicle, n = 96 FT206). (**G**) LSCC tumours stained with c-MYC, c-JUN, and Δp63 antibodies. KF animals treated with vehicle (left panel) or FT206 (right panel). Inserts showing c-MYC$^+$, c-JUN$^+$, Δp63$^+$ LSCC tumours in mice receiving vehicle (left panel) but partial positive or negative LSCC lesions in mice receiving FT206 (right panel). Scale bars, 50 μm. (**H**) Scheme depicting experimental design for in vivo test of FT206 (75 mg/kg) for 4 days consecutively (upper panel). Cleaved caspase-3 (CC3) stain shows apoptotic cells (bottom panel). Scale bars, 50 μm. (**I**) Quantification of CC3-positive cells per field (20 ×) in LADC (n = 114 vehicle, 203 FT206) and LSCC (n = 94 vehicle, 167 FT206) tumours from KF mice treated as in H. Plots indicate mean. Student's two-tailed t test was used to calculate p values. Source data for C, D, E, F, and I.

The online version of this article includes the following source data and figure supplement(s) for figure 5:

**Source data 1.** Quantification of LADC and LSCC tumours in the KF model.

**Figure supplement 1.** USP28 inhibitor FT206 tolerability in mice.

mice, USP28 inhibitor treatment was well tolerated by the experimental animals, while having a dramatic effect on LSCC regression. USP28 small-molecule inhibition phenocopies the effects of *Usp28* deletion in LSCC regression, consistent with on-target activity. However, we cannot exclude that the inhibition of USP25 and possibly additional off-targets effects may contribute to the observed phenotype. Inhibitor-treated mice kept a normal body weight, indicating no global adverse effects.

While USP28 inhibition resulted in profoundly reduced LSCC growth, the effect on LADC was modest. TP63, c-JUN, and c-MYC protein levels are increased in LSCC compared to LADC (*Figure 1C and D*). This could indicate a greater dependence of LSCC on these oncoproteins, which consequently may result in increased sensitivity to USP28 inhibition. We previously found that *Usp28* deficiency corrected the accumulation of SCF (Fbw7) substrate proteins, including c-JUN and c-MYC, in *Fbw7* mutant cells (*Diefenbacher et al., 2015*). The frequent downregulation of *FBXW7* in human LSCC (*Ruiz et al., 2019*; *Figure 1—figure supplement 2B*) may underlie the increased accumulation of SCF(Fbw7) substrate proteins like c-MYC, c-JUN, and ΔP63 in LSCC, and thereby cause LSCC tumours to be increasingly dependent on USP28 function. Indeed, our study suggests that those three oncoproteins are all relevant targets of USP28 in LSCC (*Figure 2H*). In contrast, Prieto-Garcia et al. saw no difference in c-JUN and c-MYC protein levels and suggested a different mechanism of action. Of note, our and the Prieto-Garcia et al. studies used different dual specificity inhibitors of USP28/25 that have distinct properties. FT206, the compound used in this study, preferentially inhibits USP28 compared to USP25, whereas AZ1, the compound used by Prieto-Garcia et al., showed a pronounced activity towards USP25. In addition, FT206 inhibits USP28 in the nano-molar range, while Prieto-Garcia et al. typically used AZ1 at 10–30 μM, possibly because higher compound concentrations are required for therapeutic inhibition of USP28. Therefore, differences in the selectivity and potency of the compounds used may explain some of the differences observed.

Interestingly, all human LSCC cell lines used in the xenograft experiment (*Figure 6*), each of which responded well to USP28 inhibition, do not show neither gain- or loss-of-function mutations in *USP28* nor *FBXW7*, respectively. Thus, these data support the notion that LSCC tumour cells respond to USP28 inhibition, regardless of *USP28/FBXW7* mutation status, which suggest that USP28 inhibition might be a therapeutic option for many LSCC patients.

In summary, our studies demonstrate that USP28 is a key mediator of LSCC maintenance and progression and hence USP28 represents an exciting therapeutic target. Therefore, USP28 inhibition should be considered as a potential therapy for human LSCC.

## Materials and methods
### Mice

The *Kras^LSL-G12D^* (*Jackson et al., 2001*), *Fbxw7^flox/flox^* (*Jandke et al., 2011*), *Usp28^flox/flox^* (*Diefenbacher et al., 2014*), *Kras^FSF-G12D^* (*Schönhuber et al., 2014*), *Trp53^FRT/FRT^* (*Schönhuber et al., 2014*), *Rosa26^FSF-CreERT^* (*Schönhuber et al., 2014*), *Rosa26^LSL-mTmG^* (*Muzumdar et al., 2007*) strains have been previously described. Immunocompromised NSG mice were maintained in-house. All animal experiments were

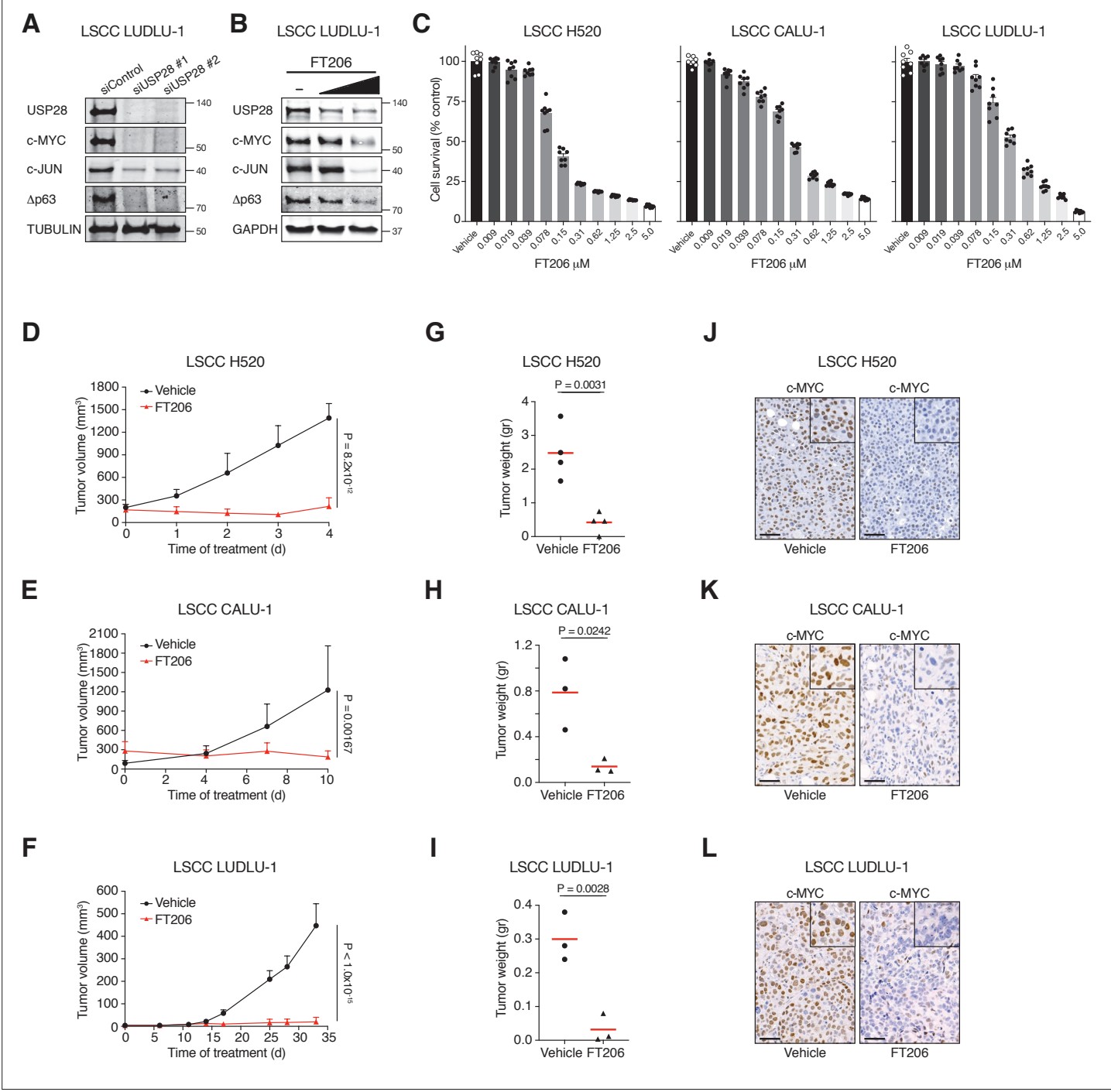

**Figure 6.** Pharmacological inhibition of ubiquitin-specific protease 28 (USP28) prevents human lung squamous cell carcinoma (LSCC) tumour progression and reduces c-MYC protein levels in xenograft models. (**A**) Small interfering RNA (siRNA)-mediated knockdown of USP28 decreases c-MYC, c-JUN, and Δp63 protein levels in human LUDLU-1 LSCC cells. (**B**) USP28 inhibition using FT206 (0.2 and 0.4 μM) reduces c-MYC, c-JUN, and Δp63 protein levels in human LUDLU-1 LSCC cells. (**C**) USP28 inhibition using FT206 decreases cell proliferation in human LSCC (NCI-H520, CALU-1, and LUDLU-1) cell lines (n = 8). Graphs indicate mean ± SEM. (**D, E, F**) In vivo tumour graft growth curves of human LSCC (NCI-H520, CALU-1, and LUDLU-1) cell lines subcutaneously injected in flanks of immunocompromised mice. Animals with palpable tumours were treated with vehicle or FT206 (75 mg/kg) via oral gavage. Plots indicate mean ± SD of the tumour volumes. p Values calculated from two-way analysis of variance (ANOVA) with Bonferroni's multiple comparisons test (NCI-H520 n = 4 vehicle and 4 FT206; CALU-1 n = 3 vehicle and 3 FT206; LUDLU-1 n = 3 vehicle and 3 FT206). (**G, H, I**) Mice treated as in **D, E, and F**, respectively. Plots showing the weight of xenograft tumours at the end point. Student's two-tailed t test was used to calculate p values (NCI-H520 n = 4 vehicle and 4 FT206; CALU-1 n = 3 vehicle and 3 FT206; LUDLU-1 n = 3 vehicle and 3 FT206). (**J, K, L**) c-MYC

*Figure 6 continued on next page*

*Figure 6 continued*

immunohistochemistry stainings of NCI-H520, CALU-1, and LUDLU-1 xenografts in mice treated as in D, E, and F, respectively. Scale bars, 50 μm. Source data for C, D, E, F, G, H, and I.

The online version of this article includes the following source data and figure supplement(s) for figure 6:

**Source data 1.** USP28 inhibition impairs tumour growth in human LSCC xenografts.

**Figure supplement 1.** USP25 deletion does not affect c-MYC,c-JUN and Δp63 protein levels.

approved by the Francis Crick Institute Animal Ethics Committee and conformed to UK Home Office regulations under the Animals (Scientific Procedures) Act 1986 including Amendment Regulations 2012. All strains were genotyped by Transnetyx. Each group contained at least three mice, which generates enough power to pick up statistically significant differences between treatments, as determined from previous experience (*Ruiz et al., 2019*). Mice were assigned to random groups before treatment.

## Generation of *Fbxw7*FRT/FRT mice

To generate a conditional allele of *Fbxw7*, we employed the CRISPR-Cas9 approach to insert two FRT sites into the intron 4 and 5 of *Fbxw7*, respectively. Two guide RNAs targeting the integration sites (gRNA-Int5A: accgtcggcacactggtcca; gRNA-Int4A: cactcgtcactgacatcgat), two homology templates containing the FRT sequences (gRNA-Int5B: agcactgacgagtgaggcgg; gRNA-Int4B: tgcctagcctttta-caagat) and the Cas9 mRNA were micro-injected into the fertilized mouse eggs. The offspring were screened by PCR and one line with proper integration of two FRT sites was identified.

## Analysis of public data from cancer genomics studies

Data from TCGA Research Network (Lung Squamous Cell Carcinoma [TCGA, Firehose Legacy]), including mutations, putative copy-number alterations, and mRNA expression (mRNA expression z-scores relative to diploid samples [RNA Seq V2 RSEM; threshold 2.0]), were analysed using cBioportal software and visualized using the standard Oncoprint output (*Cerami et al., 2012*). The Onco Query Language (OQL) used was 'USP28: MUT AMP GAIN EXP ≥ 2' 'FBXW7: MUT HOMDEL HETLOSS EXP ≤ –2'. Source data was from GDAC Firehose, previously known as TCGA Provisional. The complete sample set used was (n = 178). Expression analysis was performed using GEPIA (Gene Expression Profiling Interactive Analysis) software (2017).

## Human lung tumour analysis

Human biological samples were collected, stored, and managed by the Cordoba node belonging to the Biobank of the Andalusian Health Service (Servicio Andaluz de Salud [SAS]) and approved by the Ethics and Clinical Research Committee of the University Hospital Reina Sofia. All subjects gave informed consent. Pathologists assessed all samples before use. mRNA extracted from the samples was analysed by qPCR. Primers are listed in *Table 1*.

## Tumour induction and tamoxifen treatment

Induction of NSCLC tumours was carried out in anaesthetized (2–2.5% isoflurane) mice by intratracheal instillation of a single dose of $2.5 \times 10^7$ pfu of adenoviruses encoding either the Cre recombinase (adeno-CMV-Cre) or Flp recombinase (adeno-CMV-Flp) (*Ruiz et al., 2021*). Activation of the inducible CreERT2 recombinase was carried out by intraperitoneal injection of tamoxifen (100 μg/kg body weight) dissolved in peanut oil for 10 days.

**Table 1.** Primers for quantitative polymerase chain reaction (qPCR).

| Name | Primer (5′–3′) | |
|---|---|---|
| | Forward | Reverse |
| ACTIN | GAAAATCTGGCACCACACCT | TAGCACAGCCTGGATAGCAA |
| USP28 | ACTCAGACTATTGAACAGATGTACTGC | CTGCATGCAAGCGATAAGG |
| MYC | TCTCCTTGCAGCTGCTTAG | GTCGTAGTCGAGGTCATAG |

## CT image acquisition and processing

The SkyScan-1176, a high-resolution low-dose X-ray scanner, was used for 3D CT. Mice were anaesthetized with 2–2.5% isoflurane and CT images were acquired at a standard resolution (35 μm pixel size). The raw scan data was sorted using RespGate software, based on the position of the diaphragm, into end expiration bins. 3D reconstruction was performed using NRecon software. 3D data sets were examined using Data Viewer software.

## Mouse treatments with FT206

Nine weeks upon Ad5-CMV-Cre infection, *KRas*$^{LSL-G12D}$; *Fbxw7*$^{flox/flox}$ mice were treated with FT206 (75 mg/kg) via oral gavage on days 1, 3, and 5 per week during 5 weeks. Body weights were register every week.

## In vivo pharmacology with subcutaneous graft tumours

Human LSCC tumour cell lines (NCI-H520, CALU-1, and LUDLU-1) were resuspended as single-cell suspensions at $10^7$ cells/ml in PBS:Matrigel; 100 μl ($10^6$ cells total) of this suspension was injected into the flanks of immunodeficient NSG mice. When tumours were palpable, treatment with FT206

**Table 2.** List of reagents.

| Reagent | Source | Identifier |
| --- | --- | --- |
| Antibodies | | |
| Rabbit anti-CK5 | Abcam | Abcam Cat# ab52635, RRID:AB_869890 |
| Rabbit anti-c-MYC | Abcam | Abcam Cat# ab32072, RRID:AB_731658 |
| Goat anti-GFP | Abcam | Abcam Cat# ab6673, RRID: AB_305643 |
| Rabbit anti-Ki67 | Abcam | Abcam Cat# ab16667, RRID: AB_302459 |
| Rabbit anti-TTF1 | Abcam | Abcam Cat# ab76013, RRID:AB_1310784 |
| Rabbit anti-USP28 | Abcam | Abcam Cat# ab126604, RRID:AB_11127442 |
| Rabbit anti-USP25 | Abcam | Abcam Cat# ab187156 |
| Rabbit anti-ACTIN | Abcam | Abcam Cat# ab8227, RRID:AB_2305186 |
| Rabbit anti-USP28 | Atlas | Atlas Antibodies Cat# HPA006779, RRID:AB_1080517 |
| Rabbit anti-Δp63 | BioLegend | BioLegend Cat# 619001, RRID:AB_2256361 |
| Mouse anti-c-JUN | BD Biosciences | BD Biosciences Cat# 610326, RRID:AB_397716 |
| Rabbit anti-FBW7 | Bethyl | Bethyl Cat# A301-721A, RRID:AB_1210898 |
| Rabbit anti-USP7 | Enzo | Enzo Life Sciences Cat# BML-PW0540, RRID:AB_224147 |
| Mouse anti-GAPDH | Invitrogen | Thermo Fisher Scientific Cat# MA5-15738, RRID:AB_10977387 |
| Rabbit anti-SFTPC | Millipore | Millipore Cat# AB3786, RRID:AB_91588 |
| Rabbit anti-caspase-3 active | R&D Systems | R&D Systems Cat# AF835, RRID:AB_2243952 |
| Rat anti-HA | Roche | Roche Cat# 11666606001, RRID:AB_514506 |
| Mouse anti-TUBULIN | Sigma | Sigma-Aldrich Cat# T5168, RRID:AB_477579 |
| Mouse anti-VINCULIN | Sigma | Sigma-Aldrich Cat# V9131, RRID:AB_477629 |
| Virus strains | | |
| Adeno-CMV-Cre | UI viral vector core | VVC-U of Iowa-5-HT |
| Adeno-CMV-Flp | UI viral vector core | VVC-U of Iowa-530HT |
| Chemicals, peptides, and recombinant proteins | | |
| Doxycycline hyclate | Sigma | D9891 |
| Tamoxifen | Sigma | T5648 |

(75 mg/kg) was initiated with the same schedule on days 1, 3, and 5 per week. Tumour grafts were measured with digital callipers, and tumour volumes were determined with the following formula: (length × width$^2$) × (π/6). Tumour volumes are plotted as means ± SD.

## Histopathology, IHC, and BaseScope analysis

For histological analysis, lungs were fixed overnight in 10 % neutral buffered formalin. Fixed tissues were subsequently dehydrated and embedded in paraffin, and sections (4 μm) were prepared for H&E staining or IHC. Antibodies are given in *Table 2*. BaseScope was performed following the manufacturer's protocol. The *Usp28*-specific probe was custom-designed to target 436–482 of NM_175482.3; *Ppib* probe was used as a positive control (Bio-Techne Ltd).

Tumour numbers were counted from whole lung sections: LADC and LSCC tumours were identified by SFTPC and CK5 stains, respectively. Tumour areas (μm$^2$) were measured from lung sections using Zen3.0 (blue edition) software. For quantification of tumour cell death, the number of CC3-positive cells was counted in individual tumours per field (20 ×). The number of ΔP63$^+$, c-MYC$^+$, and c-JUN$^+$ cells was counted in individual tumours/10,000 μm$^2$. All analyses were performed uniformity across all lung sections and the whole lungs were used to derive data.

## Cell culture

Primary KF LSCC cells were cultured in N2B27 medium containing EGF (10 ng/ml; Pepro Tech) and FGF2 (20 ng/ml; Pepro Tech) (*Ruiz et al., 2019*). Human LSCC (NCI-H226, NCI-H520, CALU-1, LUDLU-1, and SKMES) and LADC (NCI-H23, NCI-H441, and NCI-H1650) lines were provided by the Francis Crick Institute Cell Services and cultured in RPMI-1640 medium supplemented with 10 % FBS, 1 % penicillin/streptomycin, 2 mM glutamine, 1 % NEEA, and 1 mM Na pyruvate. All cells were tested *Mycoplasma*-negative and maintained at 37 °C with 5 % CO$_2$.

## Cell treatments

Mouse KF LSCC and human LUDLU-1 cells were treated with vehicle or FT206 at different concentrations for 48 hr to analyse c-MYC, c-JUN, and Δp63 protein levels by Western blotting.

Primary mouse KF LSCC cells were infected with inducible shRNAs against the *Usp28* gene and then expose to doxycycline hyclate (1 μg/ml) for 48 hr. Cell number was counted using an automated cell counter (Thermo Fisher Scientific, Countess Automated Cell Counter).

Mouse KF LSCC and human cell lines were transfected with specific siRNAs against the *Myc, Jun, Tp63, Usp25,* or *Usp28* genes, using Lipofectamine RNAiMAX and 25 nM of each siRNA according to the manufacturer's instructions (Dharmacon); 48–96 hr later, cell number was counted using an automated cell counter.

For IC50, mouse KF LSCC and human cells were treated with vehicle or FT206 at different concentrations for 72 hr. Cell viability was measured as the intracellular ATP content using the CellTiter-Glo Luminescent Cell Viability Assay (Promega), following the manufacturer's instructions. IC50 was calculated using GraphPad Prism software.

## Western blot analysis

Cells were lysed in ice-cold lysis buffer (20 mM Tris HCl, pH 7.5, 5 mM MgCl$_2$, 50 mM NaF, 10 mM EDTA, 0.5 M NaCl, and 1 % Triton X-100) that was completed with protease, phosphatase, and kinase inhibitors. Protein extracts were separated on SDS-PAGE, transferred to a nitrocellulose membrane, and blotted with antibodies, which are given in *Table 2*. Primary antibodies were detected against mouse or rabbit IgGs and visualized with ECL Western blot detection solution (GE Healthcare) or Odyssey infrared imaging system (LI-COR, Biosciences).

## USP28 inhibitor synthesis

Synthesis and characterization of the USP28/25 small-molecule inhibitor FT206, a thienopyridine carboxamide derivative, has been described previously in the patent application WO 2017/139778 Al (*Guerin et al., 2017*) and more recent updates WO 2019/032863 (*Zablocki et al., 2019*) and WO 2020/033707, where FT206 is explicitly disclosed as in *Guerin et al., 2020*.

## Cellular DUB profiling using Ub-based active site directed probes

Molecular probes based on the ubiquitin scaffold were generated and used essentially as described (*Pinto-Fernández et al., 2019*; *Turnbull et al., 2017*). In brief, HA-tagged Ub propargyl probes were synthesized by expressing the fusion protein HA-Ub75-intein-chitin binding domain in *Escherichia coli* BL21 strains. Bacterial lysates were prepared, and the fusion protein was purified over a chitin binding column (NEB labs, UK). HA-Ub75-thioester was obtained by incubating the column material with mercaptosulfonate sodium salt (MESNa) overnight at 37 °C. HA-Ub75-thioester was concentrated to a concentration of ~1 mg/ml using 3000 MW filters (Sartorius) and then desalted against PBS using a PD10 column (GE Healthcare); 500 µl of 1–2 mg/ml of HA-Ub75- thioester was incubated with 0.2 mmol of bromo-ethylamine at pH 8–9 for 20 min at ambient temperature, followed by a desalting step against phosphate buffer pH 8 as described above. Ub probe material was concentrated to ~1 mg/ml, using 3000 MW filters (Sartorius), and kept as aliquots at –80 °C until use.

## DUB profiling competition assays with cell extracts and with cells

Crude NCI-H520 cell extracts were prepared as described previously using glass-bead lysis in 50 mM Tris pH 7.4, 5 mM $MgCl_2$, 0.5 mM EDTA, 250 mM sucrose, 1 mM DTT. For experiments with crude cell extracts, 50 µg of NCI-H520 cell lysate was incubated with different concentrations of USP28 inhibitor compounds (FT206 and AZ1) for 1 hr at 37 °C, followed by addition of 1 µg HA-UbPA and incubation for 10 min (*Figure 4B and C*) or 30 min (*Figure 4—figure supplement 1A* comparing FT206 and AZ1) at 37 °C. Incubation with Ub probe was optimized to minimize replacement of non-covalent inhibitor FT206 by the covalent probe. Samples were then subsequently boiled in reducing SDS-sample buffer, separated by SDS-PAGE and analysed by Western blotting using anti-HA (Roche, 1:2000), anti-USP28 (Abcam, 1:1000), anti-USP25 (Abcam, 1:1000), anti-GAPDH (Invitrogen, 1:1000), or beta Actin (Abcam, 1:2000) antibodies. For cell-based DUB profiling, $5 \times 10^6$ intact cells were incubated with different concentrations of inhibitors in cultured medium for 4 hr at 37 °C, followed by glass-bead lysis, labelling with HA-UbPA probe, separation by SDS-PAGE and Western blotting as described above.

## DUB inhibitor profiling by quantitative mass spectrometry

Ub probe pulldown experiments in presence of different concentrations of the inhibitor FT206 were performed essentially as described (*Pinto-Fernández et al., 2019*; *Turnbull et al., 2017*) with some modifications. In brief, immune precipitated material from 500 µg to 1 mg of NCI-H520 cell crude extract was subjected to in-solution trypsin digestion and desalted using C18 SepPak cartridges (Waters) based on the manufacturer's instructions. Digested samples were analysed by nano-UPLC-MS/MS using a Dionex Ultimate 3000 nano UPLC with EASY spray column (75 µm × 500 mm, 2 µm particle size, Thermo Scientific) with a 60 min gradient of 0.1 % formic acid in 5 % DMSO to 0.1 % formic acid to 35 % acetonitrile in 5 % DMSO at a flow rate of ~250 nl/min (~600 bar/40 °C column temperature). MS data was acquired with an Orbitrap Q Exactive High Field (HF) instrument in which survey scans were acquired at a resolution of 60,000 @ 400 m/z and the 20 most abundant precursors were selected for CID fragmentation. From raw MS files, peak list files were generated with MSConvert (Proteowizard V3.0.5211) using the 200 most abundant peaks/spectrum. The Mascot (V2.3, Matrix Science) search engine was used for protein identification at a false discovery rate of 1%, mass deviation of 10 ppm for MS1, and 0.06 Da (Q Exactive HF) for MS2 spectra, cys carbamidomethylation as fixed modification, met oxidation, and Gln deamidation as variable modification. Searches were performed against the UniProtKB human sequence database (retrieved 15.10.2014). Label-free quantitation was performed using MaxQuant Software (V1.5.3.8), and data further analysed using GraphPad Prism software (V7) and Microsoft Excel. Statistical test analysis of variance (ANOVA) (multiple comparison; original FRD method of Benjamini and Hochberg) was performed using GraphPad Prism software. The MS data was submitted to PRIDE for public repository with an internal ID of px-submission #469830.

## TUBE pulldown

Endogenous poly-Ub conjugates were purified from cells using TUBE affinity reagents (LifeSensors, UM401). Cells were lysed in buffer containing 50 mM Tris-HCl pH 7.5, 0.15 M NaCl, 1 mM EDTA, 1% NP-40, 10 % glycerol supplemented with complete protease inhibitor cocktail, PR-619, and 1,10-phenanthroline. Lysate was cleared by centrifugation, Agarose-TUBEs were added, and

pulldown was performed for 16 hr at 4 °C on rotation. The beads were then washed three times with 1 ml of ice-cold TBS-T, and bound material was eluted by mixing the beads with sample buffer and heating to 95 °C for 5 min.

## Statistical analysis

Data are represented as mean ± SEM. Statistical significance was calculated with the unpaired two-tailed Student's t test, one-way or two-way ANOVA followed by multiple comparison test using GraphPad Prism software. A p value that was less than 0.05 was considered to be statistically significant for all data sets. Significant differences between experimental groups were: *$p < 0.05$, **$p < 0.01$, or ***$p < 0.001$. Biological replicates represent experiments performed on samples from separate biological preparations; technical replicates represent samples from the same biological preparation run in parallel.

## Acknowledgements

Part of this work was funded by Forma Therapeutics. This work was also supported by the Francis Crick Institute which receives its core funding from Cancer Research UK (FC001039), the UK Medical Research Council (FC001039), and the Wellcome Trust (FC001039). We thank the Discovery Proteomics Facility (led by Dr Roman Fischer) at the Target Discovery Institute (Oxford) for expert help with the analysis by mass spectrometry. Work in the BMK laboratory was supported by a John Fell Fund 133/075, the Wellcome Trust (097813/Z/11/Z), and the Engineering and Physical Sciences Research Council (EP/N034295/1).

## Additional information

### Competing interests

Andrew P Turnbull: Andrew P Turnbull is affiliated with the CRUK Therapeutic Discovery Laboratories at the Crick Institute, for which no financial interests have been declared. APT declares competing financial interests due to financial support for the project described in this manuscript by Forma Therapeutics, Watertown, MA, USA.. Wojciech W Krajewski: Wojciech W Krajewski is affiliated with the CRUK Therapeutic Discovery Laboratories at the Crick Institute, for which no financial interests have been declared. WWK declares competing financial interests due to financial support for the project described in this manuscript by Forma Therapeutics, Watertown, MA, USA.. David Guerin: Dave Guerin is affiliated with Constellation Pharmaceuticals (Boston, USA), for which no financial interests have been declared. DG declares competing financial interests due to financial support for the project described in this manuscript by Forma Therapeutics, Watertown, MA, USA.. Jeffrey D Kearns: Jeffrey Kearns is affiliated with the Novartis Institutes for BioMedical Research (Boston, USA), for which no financial interests have been declared. JK declares competing financial interests due to financial support for the project described in this manuscript by Forma Therapeutics, Watertown, MA, USA.. Stephanos Ioannidis: Stephanos Ioannidis is affiliated with H3 Biomedicine (Cambridge, MA, USA), for which no financial interests have been declared. SI declares competing financial interests due to financial support for the project described in this manuscript by Forma Therapeutics, Watertown, MA, USA.. Marie Katz: Marie Katz is affiliated with Valo Health (Boston, USA), for which no financial interests have been declared. MK declares competing financial interests due to financial support for the project described in this manuscript by Forma Therapeutics, Watertown, MA, USA.. Crystal McKinnon: Crystal McKinnon is affiliated with Valo Health (Boston, USA), for which no financial interests have been declared. CM declares competing financial interests due to financial support for the project described in this manuscript by Forma Therapeutics, Watertown, MA, USA.. Jonathan O'Connell: Johnathan O'Connell is affiliated with Valo Health (Boston, USA), for which no financial interests have been declared. JOC declares competing financial interests due to financial support for the project described in this manuscript by Forma Therapeutics, Watertown, MA, USA.. Neil Jones: Neil Jones is affiliated with the CRUK Therapeutic Discovery Laboratories at the Crick Institute, for which no financial interests have been declared. NJ declares competing financial interests due to financial support for the project described in this manuscript by Forma Therapeutics, Watertown, MA,

USA.. Claire Heride: Claire Heride is affiliated with the CRUK Therapeutic Discovery Laboratories at the Crick Institute, for which no financial interests have been declared. CH declares competing financial interests due to financial support for the project described in this manuscript by Forma Therapeutics, Watertown, MA, USA.. Min Wu: Min Wu is affiliated with Disc Medicine (Cambridge, MA, USA), for which no financial interests have been declared. MW declares competing financial interests due to financial support for the project described in this manuscript by Forma Therapeutics, Watertown, MA, USA.. Christopher J Dinsmore: Christopher J Dinsmore is affiliated with Disc Medicine (Cambridge, MA, USA), for which no financial interests have been declared. CJD declares competing financial interests due to financial support for the project described in this manuscript by Forma Therapeutics, Watertown, MA, USA.. Tim R Hammonds: Tim R Hammonds is affiliated with the CRUK Therapeutic Discovery Laboratories at the Crick Institute, for which no financial interests have been declared. TRH declares competing financial interests due to financial support for the project described in this manuscript by Forma Therapeutics, Watertown, MA, USA.. Sunkyu Kim: Sunkyu Kim is affiliated with Incyte (Wilmington, DE, USA), for which no financial interests have been declared. SK declares competing financial interests due to financial support for the project described in this manuscript by Forma Therapeutics, Watertown, MA, USA.. David Komander: DK declares competing financial interests due to financial support for the project described in this manuscript by Forma Therapeutics, Watertown, MA, USA.. Sylvie Urbe: SU declares competing financial interests due to financial support for the project described in this manuscript by Forma Therapeutics, Watertown, MA, USA.. Michael J Clague: MJC declares competing financial interests due to financial support for the project described in this manuscript by Forma Therapeutics, Watertown, MA, USA.. Benedikt M Kessler: BMK declares competing financial interests due to financial support for the project described in this manuscript by Forma Therapeutics, Watertown, MA, USA.. Axel Behrens: AB declares competing financial interests due to financial support for the project described in this manuscript by Forma Therapeutics, Watertown, MA, USA.. The other authors declare that no competing interests exist.

## Funding

| Funder | Grant reference number | Author |
|---|---|---|
| Forma Therapeutics | 2013-2019 | E Josue Ruiz<br>Adan Pinto-Fernandez<br>Andrew P Turnbull<br>Linxiang Lan<br>Thomas M Charlton<br>Hannah C Scott<br>Andreas Damianou<br>George Vere<br>Eva M Riising<br>Clive Da Costa<br>Wojciech W Krajewski<br>David Guerin<br>Jeffrey D Kearns<br>Stephanos Ioannidis<br>Marie Katz<br>Crystal McKinnon<br>Jonathan O'Connell<br>Natalia Moncaut<br>Ian Rosewell<br>Emma Nye<br>Neil Jones<br>Claire Heride<br>Malte Gersch<br>Min Wu<br>Christopher J Dinsmore<br>Tim R Hammonds<br>Sunkyu Kim<br>David Komander<br>Sylvie Urbe<br>Michael J Clague<br>Benedikt M Kessler<br>Axel Behrens |
| Cancer Research UK Manchester Centre | FC001039 | Axel Behrens |

| Funder | Grant reference number | Author |
|---|---|---|
| Medical Research Council | FC001039 | Axel Behrens |
| Wellcome Trust | FC001039 | Axel Behrens |
| Wellcome Trust | 097813/Z/11/Z | Benedikt M Kessler |
| Engineering and Physical Sciences Research Council | EP/N034295/1 | Benedikt M Kessler |
| John Fell Fund, University of Oxford | 133/075 | Benedikt M Kessler |

The funders had no role in study design, data collection and interpretation, or the decision to submit the work for publication.

## Author contributions

E Josue Ruiz, Data curation, Formal analysis, In vivo mouse tumour work, Investigation, Methodology, Validation, Visualization, Writing – original draft, Writing – review and editing; Adan Pinto-Fernandez, Data curation, Formal analysis, Investigation, Methodology, Validation, Visualization, Writing – review and editing; Andrew P Turnbull, Formal analysis, Investigation, Validation; Linxiang Lan, Investigation, Methodology, Validation, Visualization; Thomas M Charlton, Hannah C Scott, Andreas Damianou, George Vere, Eva M Riising, Wojciech W Krajewski, Ian Rosewell, Emma Nye, Claire Heride, Malte Gersch, Investigation, Methodology; Clive Da Costa, Natalia Moncaut, Investigation, Methodology, Visualization; David Guerin, Funding acquisition, Methodology, Project administration, Resources, Supervision; Jeffrey D Kearns, Crystal McKinnon, Jonathan O'Connell, Project administration, Resources, Supervision; Stephanos Ioannidis, Conceptualization, Project administration, Resources, Supervision; Marie Katz, Resources, Supervision; Neil Jones, Conceptualization, Funding acquisition, Resources, Supervision; Min Wu, Project administration, Supervision; Christopher J Dinsmore, Sylvie Urbe, Michael J Clague, Conceptualization, Funding acquisition, Project administration, Supervision; Tim R Hammonds, Sunkyu Kim, Conceptualization, Funding acquisition, Supervision; David Komander, Conceptualization, Investigation, Project administration, Supervision; Benedikt M Kessler, Conceptualization, Funding acquisition, Supervision, Writing – original draft, Writing – review and editing; Axel Behrens, Conceptualization, Funding acquisition, Project administration, Supervision, Writing – original draft, Writing – review and editing

## Author ORCIDs

Adan Pinto-Fernandez (iD) http://orcid.org/0000-0003-1693-9664
George Vere (iD) http://orcid.org/0000-0002-1154-5212
Jeffrey D Kearns (iD) http://orcid.org/0000-0001-7261-5096
Claire Heride (iD) http://orcid.org/0000-0003-1927-9577
Michael J Clague (iD) http://orcid.org/0000-0003-3355-9479
Benedikt M Kessler (iD) http://orcid.org/0000-0002-8160-2446
Axel Behrens (iD) http://orcid.org/0000-0002-1557-1143

## Ethics

All animal experiments were approved by the Francis Crick Institute Animal Ethics Committee and conformed to UK Home Office regulations under the Animals (Scientific Procedures) Act 1986 including Amendment Regulations 2012.

## Decision letter and Author response

Decision letter https://doi.org/10.7554/eLife.71596.sa1
Author response https://doi.org/10.7554/eLife.71596.sa2

# Additional files

## Supplementary files

- Transparent reporting form
- Source data 1. Complete immunblots and gel figures.

## Data availability

For all figures with graphs we provide source data files in the Supplemental Information section. The mass spectrometry proteomics data have been deposited to the ProteomeXchange Consortium via the PRIDE partner repository with an internal ID of PXD469830.

The following dataset was generated:

| Author(s) | Year | Dataset title | Dataset URL | Database and Identifier |
|---|---|---|---|---|
| Ruiz E | 2021 | Data set title: USP28 deletion and small molecule inhibition destabilises c-Myc and elicits regression of squamous cell lung carcinoma | https://www.ebi.ac.uk/pride/archive/projects/PXD469830 | PRIDE, PXD469830 |

The following previously published datasets were used:

| Author(s) | Year | Dataset title | Dataset URL | Database and Identifier |
|---|---|---|---|---|
| Hammermann et al | 2012 | Genetic alterations in USP28 gene in human LSCC | http://gdac.broadinstitute.org/runs/stddata__2016_01_28/data/LUSC/20160128/ | LSCC TCGA data, 20160128 |

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
