## [Decision Letter]

**Acceptance summary:**

This study identifies a specific requirement for ubiquitin-specific protease 28 (USP28) in the survival of lung squamous cell carcinomas (LSCCs). Using genetic knockout models and a novel small molecule inhibitor of USP28, they demonstrate loss or inhibition of USP28 impairs the growth of LSCCs to a much greater degree than lung adenocarcinomas, associated with a greater degree of loss of c-MYC, c-JUN, and Δp63. This work will be of interest to researchers seeking to better understand and treat LSCCs.

**Decision letter after peer review:**

[Editors’ note: the authors submitted for reconsideration following the decision after peer review. What follows is the decision letter after the first round of review.]

Thank you for submitting your work entitled "USP28 deletion and small molecule inhibition destabilises c-Myc and elicits regression of squamous cell lung carcinoma" for consideration by *eLife*. Your article has been reviewed by 3 peer reviewers, one of whom is a member of our Board of Reviewing Editors, and the evaluation has been overseen by a Senior Editor. The following individual involved in review of your submission has agreed to reveal their identity: Wee-Wei Tee (Reviewer #3).

*Reviewer #1:*

The authors investigate a role for a candidate new inhibitor of USP28 in destabilizing c-MYC to reduce the growth of lung squamous carcinomas. They demonstrate that c-MYC levels are higher in lung squamous cell carcinomas (LSCC) versus lung adenocarcinomas (LADC), and depletion of c-MYC reduces LSCC cell growth. The deubiquitinase USP28 is known to stabilize c-MYC; the authors show that depletion of USP28 also decreases c-MYC protein levels. USP28 action opposes that of a ubiquitin complex targeted by the FBXW7 tumor suppressor; the authors create a new mouse model in which FLP recombinase initially causes deletion of FBXW7 and activation of KRAS to cause tumorigenesis with LSCC and LADC, followed by tamoxifen-dependent CRE recombinase deletion of USP28. Loss of USP28 in this model reduced numbers of LSCC but not LADC, and led to decreased expression of c-MYC and other short-lived proteins such as c-JUN and deltap63. A limitation of the data shown is that tumor number calculations are shown for a relatively small number of mice. Deletion of USP28 also did not restrict LADC growth in a second mouse model, with tumors forming based on activation of KRAS and loss of TP53. The authors then describe a compound, FT206, which they show is a specific inhibitor of USP28 among other ubiquitinases. They demonstrate that this compound reduces expression of c-MYC, c-JUN, and deltap63. They also show FT206 reduces growth of LSCC but not LADC in the KRAS/FBXW7 tumor model, and in human LSCC xenografts. These latter data suggest the compound FT206 may be useful as a lead compound. However, the current data are not sufficient to demonstrate FT206 binding and biological effect is specific for USP28, as the compound may also bind and regulate other non-deubiquitinase proteins.

1. Given the article focuses on c-MYC, abruptly introducing c-Jun and deltap63 as also elevated in LSCC is abrupt and seems off topic except as a way of bridging to USP28. This connects directly to a cited recent paper by Prieto-Garcia that implicates deltap63 as the critical target of a USP28 inhibitor. This makes the focus on c-MYC difficult to follow.

2. The data showing specificity for LSCC in Figure 2 appear to be based on a total of 4-5 mice/genotype for the +/- tamoxifen groups, and the number of tumors for LADC is extremely heterogeneous between animals. If you look at main figure 2C-F, and Supp Figure S3, you can see the vehicle and Tam mice are the same samples. Oddly, the statistical significance differs between the two figures. There are only 3 LSCC mice treated with Tam+FT in Supp Figure S3 – the very small number of mice could itself be a reason there isn't a significant difference with TAM, as the numbers of tumors seem to actually be trending lower. It is necessary to add additional mice to the study.

3. FT206 is a critical reagent for the study. While data shown indicates it is specific for USP28 among USPs (although with partial activity for USP25), it provides no evidence to show that it does not bind non-USP targets. Many compounds bind strongly to multiple targets. MYC expression is highly sensitive to transcriptional inhibitors; BET inhibitors, for example, also result in a rapid loss of MYC expression. It is essential to show data to establish whether this compound inhibits cell or tumor growth, and reduces MYC levels, in a USP28-/- null or USP28 depleted background, to determine whether the biological effect is related to USP28 or another potential target. It would also be important to do parallel IC50 curve determination and measurement of MYC expression in human LSCC and LADC cell lines, to supplement Figure 6 and provide support for the idea the compound acts differently in these cell models.

4. The design and findings of the study are very similar to that described by Prieto-Garcia; in that case, deltaNp63 was identified as the primary target of USP28. The discussion should explicitly discuss the relation of the present study to that published work, to emphasize points of similarity and difference.

*Reviewer #2:*

In this work Ruiz et al., use a couple of elegant mouse genetic models – KFCU (Fbxw7 deletion and mutant Ras over-expression) and KPCU (p53 deletion and mutant Ras over-expression) – to generate both LADC and LSCC tumors. Using this system, the authors show that deletion of USP28 resulted in less LSCC but not LADC tumor formation. However, both tumor types showed overall decrease in tumor size (in KFCU not shown in KPCU). These results are the genetic proof of concept that USP28 inhibition will be particularly detrimental in the context of LSCC tumors. They further test a compound (FT206) that was previously found to target USP28 and show that indeed this compound is specific for USP28 binding among USPs and can reduce the tumor numbers and size only in LSCC tumors and not LADC in the KF model and in three separate LSCC cell line xenograft models. Altogether making the argument that targeting LSCC tumors with chemical inhibitors of USP28 is a promising clinical strategy for LSCC cancers. Overall this paper is interesting and the results provided in vivo are very strong and nicely demonstrate an on-target effect of FT206 and its specificity in LSCC tumors. The work is very similar to a recent publication of (Prieto-Garcia EMBO Mol Med 2020) describing very similar results for USP28 dependency in LSCC tumors and previous findings regarding the chemical matter used in this paper (FT206).

The major strengths of this paper is that the authors use several very elegant mouse models to establish that Usp28 is a good candidate target for potential therapeutic development designated for LSCC patients. They also show the proof of concept using a compound that is described as a Usp28 inhibitor (FT206). It should be noted that much of the genetic data, showing the importance of Usp28 in LSCC was previously described (Prieto-Garcia EMBO Mol Med 2020) including the potential benefit of chemical inhibition of USP28. A potential weakness is that there is no rigorous characterizing of Usp28 substrate ubiquitination and degradation following FT206 treatment. This work will likely motivate the development of the USP28 inhibitor(s) for further preclinical assessment in Usp28 dependent tumors such as LSCC.

1. It is not clear in the paper as it is written now how the authors selected FT206. Is this a novel compound? Or was its function as a USP28 inhibitor already described before? There is a couple of publication cited and additional patents but even going through them it does not clarify this point. As this is a central novelty of this paper It would be informative to give a proper background on this compound, why was it selected? Was there a previous screen conducted ? is it a modification of a previously described scaffold ? etc., It makes no sense to dig through the patents to try and figure this out.

2. Despite the nice results in vivo and competition assays using ABPP there is no evidence provided to show that there is actually increased ubiquitination and degradation of the substrate proteins following FT206 treatment in cells. If this is the first demonstration of this USP28 inhibitor one would want to show at least in cell culture substrate ubiquitination and degradation. Also, there seems to be consistent decrease in Usp28 levels following FT206, this needs to be addressed in the text.

*Reviewer #3:*

The prevalent treatment options for LSCC are limited in efficacy. Through genetic inactivation of Usp28 in a novel lung cancer mouse model, and chemical inhibition of Usp28 in induced LSCC in mice and human LSCC xenograft tumors, the authors demonstrated the specific dependency of LSCC (but not LADC) on the protein deubiquitinase Usp28. The authors also showed that loss of Usp28 by either means leads to depletion of the oncoproteins c-Myc, p63 and c-Jun in LSCC. Finally, the authors described a novel small molecule that is specific for Usp25/28. Based on these results, the authors suggested chemically targeting USP28 as a potential therapeutic option for human LSCC patients.

Strengths: The presentation of the work is clear, concise and easily readable. The data presented largely supports the authors' conclusions on the role of USP28 in LSCC tumorigenesis and that inhibition of USP28 is a viable therapeutic option for LSCC treatment. The generation of the KFCU mice model that can give rise to both LADC and LSCC concurrently is interesting and presents a valuable tool for the wider cancer community.

Weakness: The manuscript can benefit from a deeper analysis of the relationship between FBW7 and USP28 in patient cohorts. A comparison of the activity/efficacy of FT206 to existing USP28 inhibitors will also be helpful.

1. The authors mentioned that 25% of human lung squamous cell carcinoma cases show gain of function alterations in USP28, based on TCGA data.

– What is the proportion of cases in the current study cohort (n=17) which show similar gain of function alterations at the DNA level, as well as overexpression of USP28 at the protein level (by immunohistochemistry or immunoblotting)?

– What is the correlation between DNA alterations and mRNA/protein expression? This would be clinically relevant if USP28 inhibitors are to be used in clinic, since we need a robust predictive test to select for patients who are most likely to respond to this therapy.

2. A previous study by the same first author mentioned that 69% of patients with LSCC show loss of FBW7 expression by immunohistochemistry.

– What is the FBW7 status of the cohort in the current study?

– Can USP28 overexpression/gain of function co-exists with FBW7 loss, or are they mutually exclusive?

– Apart from USP28 gain of function, FBW7 loss of function may also predict sensitivity to USP28 inhibition. Related to this, what is the status of FBW7 and USP28 in the three human LSCC cell lines used in the xenograft studies? This would clarify if the observed effect for USP28 inhibition is specific to LSCC cell lines with USP28 overexpression/FBW7 loss or to LSCC cell lines in general, regardless of USP28/FBW7 status.

3. It is interesting that LADC is not affected by the loss of Usp28. What is c-MYC and c-Jun protein expression in the LADC lesions in the KFCU mice? Are they upregulated upon KrasG12D activation and Fbw7 deletion? Although the authors showed in Figure 1 that the expression of c-MYC is lower in LADC compared to LSCC, it will be important to directly assess whether loss of USP28 (either by siRNA knockdown or FT206 treatment) in LADC cell lines can affect c-MYC protein expression.

4. As an important part of the paper is about application of the new Usp28 inhibitor FT206, the authors should have compared the efficacy with previously described Usp25/28 inhibitor (Wrigley et al; 2017, ACS Chem Biol 12,3113-3125) or at the very least, comment on potential similarities/differences/improvements etc.

5. As the authors advocate targeting USP28 in LSCC treatment, have they tested, or can they comment on, whether USP28 inhibition would be beneficial for LSCC that do not have gain-of-function alterations in USP28 (which represent 75% of LSCC)?

[Editors’ note: further revisions were suggested prior to acceptance, as described below.]

Thank you for resubmitting your work entitled "USP28 deletion and small molecule inhibition destabilises c-Myc and elicits regression of squamous cell lung carcinoma" for further consideration by *eLife*. Your revised article has been evaluated by Erica Golemis (Senior Editor) and a Reviewing Editor.

The manuscript has been improved but there are some minor remaining issues that need to be addressed, as outlined below. Please address these points.

*Reviewer #1:*

This is a revised submission of a manuscript initially deemed as of interest, but rejected due to the need for a large number of experiments to validate the author's model. In this revised submission, the authors have substantially addressed essentially every point made by the referees, providing new in vitro and in vivo data. The work now convincingly makes the point that the ubiquitin protease USP28 is specifically required to support the growth of lung squamous cell carcinomas (LSCCs), more than lung adenocarcinomas (LADCs). The work defines critical USP28 targets as c-Myc, c-Jun, and ∆p63, and shows differential USP28-dependent stability of these proteins in LSCCs and LADCs. The authors also show that the FT206 compound inhibits USP28, and is selectively effective in reducing the growth of LSCCs versus LADCs. Overall, the data provided are much improved, as is the discussion of the context of the work; and the finding is potentially clinically important.

*Reviewer #2:*

Overall the authors have addressed most of my concerns in the previous submission. I find the revised manuscript much clearer and significantly improved. I only have 4 additional points for the authors' clarification:

1) In light of the new data showing that FT206 can also inhibit USP25, albeit to a lesser extent than USP28, I think it is important to address if depletion of USP25 also results in loss of c-Myc, c-Jun and ∆p63 expressions compared to USP28 knockdown, at least in cell lines. Related to this, what is the phenotype of USP25 KO mice (if known)?

2) The authors mentioned that inhibitor treated mice kept a normal body weight, indicating no global adverse effects (Figure S5A). The numbers of mice used (n = 3) are on the low side. Do the authors have body weight and survival information for the other FT206 treated genetic mouse models (e.g Figure 2)?

3) The authors showed in Figure S2A that there is a positive correlation between USP28 copy number gain and higher mRNA expression in human LSCC patients, which may be useful for patient selection. However, they also mentioned that the 3 LSCC cell lines used for the xenograft studies do not show gain-of-function mutations in USP28 but responded well to USP28 inhibition. Do the authors have any data comparing the effects of USP28 deletion and/or inhibition in LSCC cell lines with or without USP28 alterations?

4) Related to point 3, can the authors speculate or discuss why USP28 deletion/inactivation has a less pronounced effect in LADC despite the latter also having high expression of USP28 and c-MYC ?

---

## [Author Response]

[Editors’ note: the authors resubmitted a revised version of the paper for consideration. What follows is the authors’ response to the first round of review.]

Reviewer #1:The authors investigate a role for a candidate new inhibitor of USP28 in destabilizing c-MYC to reduce the growth of lung squamous carcinomas. They demonstrate that c-MYC levels are higher in lung squamous cell carcinomas (LSCC) versus lung adenocarcinomas (LADC), and depletion of c-MYC reduces LSCC cell growth. The deubiquitinase USP28 is known to stabilize c-MYC; the authors show that depletion of USP28 also decreases c-MYC protein levels. USP28 action opposes that of a ubiquitin complex targeted by the FBXW7 tumor suppressor; the authors create a new mouse model in which FLP recombinase initially causes deletion of FBXW7 and activation of KRAS to cause tumorigenesis with LSCC and LADC, followed by tamoxifen-dependent CRE recombinase deletion of USP28. Loss of USP28 in this model reduced numbers of LSCC but not LADC, and led to decreased expression of c-MYC and other short-lived proteins such as c-JUN and deltap63. A limitation of the data shown is that tumor number calculations are shown for a relatively small number of mice. Deletion of USP28 also did not restrict LADC growth in a second mouse model, with tumors forming based on activation of KRAS and loss of TP53. The authors then describe a compound, FT206, which they show is a specific inhibitor of USP28 among other ubiquitinases. They demonstrate that this compound reduces expression of c-MYC, c-JUN, and deltap63. They also show FT206 reduces growth of LSCC but not LADC in the KRAS/FBXW7 tumor model, and in human LSCC xenografts. These latter data suggest the compound FT206 may be useful as a lead compound. However, the current data are not sufficient to demonstrate FT206 binding and biological effect is specific for USP28, as the compound may also bind and regulate other non-deubiquitinase proteins.1. Given the article focuses on c-MYC, abruptly introducing c-Jun and deltap63 as also elevated in LSCC is abrupt and seems off topic except as a way of bridging to USP28. This connects directly to a cited recent paper by Prieto-Garcia that implicates deltap63 as the critical target of a USP28 inhibitor. This makes the focus on c-MYC difficult to follow.

We agree that the clarity of our manuscript needed to be improved. To clarify the relative importance of c-Jun and ∆p63 as USP28 substrates in LSCC, we have targeted c-Jun and p63 expression by siRNAs in LSCC cells. Our new data suggest that also c-Jun and p63 are required for efficient cell proliferation in human LSCC cell lines (new Figure S1). Therefore, the inhibition of LSCC growth by USP28 inhibition appears to be due to a combination of effects on 3 independent USP28 substrates, c-Myc, c-Jun and ∆p63.

2. The data showing specificity for LSCC in Figure 2 appear to be based on a total of 4-5 mice/genotype for the +/- tamoxifen groups, and the number of tumors for LADC is extremely heterogeneous between animals. If you look at main figure 2C-F, and Supp Figure S3, you can see the vehicle and Tam mice are the same samples. Oddly, the statistical significance differs between the two figures. There are only 3 LSCC mice treated with Tam+FT in Supp Figure S3 – the very small number of mice could itself be a reason there isn't a significant difference with TAM, as the numbers of tumors seem to actually be trending lower. It is necessary to add additional mice to the study.

This is an important point. Firstly, we want to clarify that the vehicle- and Tamoxifen-treated mice indeed are the same in Figures 2 and S3. We split the data in two Figures because FT206 treatment was mentioned in the manuscript text later (Figure 5), and we only used the vehicle and USP28 KO data in Figure 2. Regarding the statistical analysis, in Figure 2, we are comparing just 2 groups, for which we used a T-test. In Figure S3 we are comparing 3 groups, thus we could not use a T-test. Instead, we used One-way ANOVA. However, to avoid misinterpretations, in the revised version of our manuscript, we show the 3 groups (vehicle- and Tamoxifen-treated and FT206-treated) in Figure 2.

As requested, we have increased the number of mice for all 3 experimental groups. We increased the number of mice in the Vehicle cohort from 5 to 8, the number of mice in the Tamoxifen-treated cohort from 4 to 7 and the number of mice in the Tamoxifen+FT206-treated cohort from 3 to 7. Consistent with our previous findings, KFCU mice exposed to tamoxifen to delete the conditional *Usp28* floxed alleles, showed a significant reduction in the numbers of LSCC lesions (new Figure 2F). In contrast, loss of Usp28 did not reduce the number of LADC tumors (new Figure 2C).

In addition, KFCU mice pre-exposed to tamoxifen, and therefore homozygous deleted for both *usp28* alleles, were treated with the USP28 inhibitor FT206. Usp28 inhibition did not result in a further reduction of LADC and LSCC lesions (new Figure 2C,D,F,G), suggesting that FT206 targets specifically Usp28.

3. FT206 is a critical reagent for the study. While data shown indicates it is specific for USP28 among USPs (although with partial activity for USP25), it provides no evidence to show that it does not bind non-USP targets. Many compounds bind strongly to multiple targets. MYC expression is highly sensitive to transcriptional inhibitors; BET inhibitors, for example, also result in a rapid loss of MYC expression. It is essential to show data to establish whether this compound inhibits cell or tumor growth, and reduces MYC levels, in a USP28-/- null or USP28 depleted background, to determine whether the biological effect is related to USP28 or another potential target. It would also be important to do parallel IC50 curve determination and measurement of MYC expression in human LSCC and LADC cell lines, to supplement Figure 6 and provide support for the idea the compound acts differently in these cell models.

To address the specificity of FT206 towards USP28, we used siRNAs to knockdown Usp28 in LSCC cells, followed by FT206 treatment. Our data showed that in a USP28-depleted background, FT206 neither affected cell growth nor reduced c-Myc protein levels (new Figure S4D). Thus, in line with our observations in the KFCU mouse model, see point 2 above, also this data strongly suggests that the effects of FT206 are mediated by Usp28 (new Figure 2C,D,F,G).

As requested, human LADC and LSCC cell lines were treated with FT206 (IC50 doses) to analyse its effect on c-Myc protein levels (new Figure S6).

Interestingly, FT206 treatment reduced c-Myc (>65%) and c-Jun (>80%) protein levels in the three human LSCC cell lines used in the xenograft studies (new Figure S6A).

Usp28 inhibition also decreased c-Myc levels in 2/3 LADC lines, although the reduction in cMyc protein levels appear to be significantly less pronounced than observed in LSCC (~60%) (new Figure S6B). In contrast, treatment of LADC cells with FT206 resulted in a significant increase in c-Jun protein levels. Thus, Usp28 inhibition acts differently in LADC and LSCC tumor cells.

4. The design and findings of the study are very similar to that described by Prieto-Garcia; in that case, deltaNp63 was identified as the primary target of USP28. The discussion should explicitly discuss the relation of the present study to that published work, to emphasize points of similarity and difference.

We thank the reviewer for this important suggestion. We have addressed the differences of our work with the study of Prieto-Garcia et al. in the Discussion section. There are key differences between our study as compared to the study of Prieto-Garcia et al. We show that c-Jun, p63 and c-Myc are all relevant targets of Usp28 in LSCC and contribute to the observed effects of Usp28 inhibition/inactivation (Figure 2H), whereas Prieto-Garcia et al. see no difference in c-Jun and c-Myc protein levels (KPLU is the LSCC model with Usp28 inactivation). We would like to point out that in our view the technical quality of our Western blot experiments is higher. We also want to emphasise that both c-Jun and c-Myc are well established targets of Usp28, and that in this revision we show ubiquitylation assays clearly demonstrating increased c-Myc ubiquitylation in the presence of Usp28 inhibitor (new Figure S4B). Therefore, the proposed mechanism of action between our study and the study by

Prieto-Garcia et al. is different.

Of note, our and the Prieto-Garcia et al. studies used different dual specificity inhibitors of Usp25/Usp28 that have distinct properties. Firstly, FT206, the compound used in our study, preferentially inhibits Usp28 compared to Usp25, whereas AZ1, the compound used by PrietoGarcia et al., showed pronounced activity towards Usp25 (new Figure S4A of the revised manuscript). Secondly, whereas FT206 inhibits Usp28 in the nano-molar range (e.g see Figure 6B and new Figure S6A of the revised manuscript), Prieto-Garcia et al. typically used AZ1 at 10-30µM. We observed a clear reduction of c-Myc levels at 300nM (new Figure S6A of the revised manuscript). Therefore, differences in the selectivity and potency of the compounds used may explain some of the differences observed. We now discuss these differences in the revised manuscript.

Reviewer #2:In this work Ruiz et al., use a couple of elegant mouse genetic models – KFCU (Fbxw7 deletion and mutant Ras over-expression) and KPCU (p53 deletion and mutant Ras over-expression) – to generate both LADC and LSCC tumors. Using this system, the authors show that deletion of USP28 resulted in less LSCC but not LADC tumor formation. However, both tumor types showed overall decrease in tumor size (in KFCU not shown in KPCU). These results are the genetic proof of concept that USP28 inhibition will be particularly detrimental in the context of LSCC tumors. They further test a compound (FT206) that was previously found to target USP28 and show that indeed this compound is specific for USP28 binding among USPs and can reduce the tumor numbers and size only in LSCC tumors and not LADC in the KF model and in three separate LSCC cell line xenograft models. Altogether making the argument that targeting LSCC tumors with chemical inhibitors of USP28 is a promising clinical strategy for LSCC cancers. Overall this paper is interesting and the results provided in vivo are very strong and nicely demonstrate an on-target effect of FT206 and its specificity in LSCC tumors. The work is very similar to a recent publication of (Prieto-Garcia EMBO Mol Med 2020) describing very similar results for USP28 dependency in LSCC tumors and previous findings regarding the chemical matter used in this paper (FT206).The major strengths of this paper is that the authors use several very elegant mouse models to establish that Usp28 is a good candidate target for potential therapeutic development designated for LSCC patients. They also show the proof of concept using a compound that is described as a Usp28 inhibitor (FT206). It should be noted that much of the genetic data, showing the importance of Usp28 in LSCC was previously described (Prieto-Garcia EMBO Mol Med 2020) including the potential benefit of chemical inhibition of USP28. A potential weakness is that there is no rigorous characterizing of Usp28 substrate ubiquitination and degradation following FT206 treatment. This work will likely motivate the development of the USP28 inhibitor(s) for further preclinical assessment in Usp28 dependent tumors such as LSCC.1. It is not clear in the paper as it is written now how the authors selected FT206. Is this a novel compound? Or was its function as a USP28 inhibitor already described before? There is a couple of publication cited and additional patents but even going through them it does not clarify this point. As this is a central novelty of this paper It would be informative to give a proper background on this compound, why was it selected? Was there a previous screen conducted ? is it a modification of a previously described scaffold ? etc., It makes no sense to dig through the patents to try and figure this out.

We thank the reviewer for this comment. In fact, the small molecule compound FT206 is part of a patent on USP28 inhibitors that was filed by Forma Therapeutics WO 2020/033707 (Guerin D et al., 2020 – referenced in our manuscript), in which several hundred compound derivatives were reported with various degrees of inhibitory potency towards USP28. Amongst those, FT206, explicitly disclosed as compound example 11.1, has been specifically optimised for in vivo use with an adequate stability in serum, suitable biodistribution and pharmacodynamics (DMPK) as described in the patent. We have added a paragraph in the result section that mentions these characteristics (page 9, lines 3-7).

2. Despite the nice results in vivo and competition assays using ABPP there is no evidence provided to show that there is actually increased ubiquitination and degradation of the substrate proteins following FT206 treatment in cells. If this is the first demonstration of this USP28 inhibitor one would want to show at least in cell culture substrate ubiquitination and degradation. Also, there seems to be consistent decrease in Usp28 levels following FT206, this needs to be addressed in the text.

As suggested, we performed ubiquitination assays on c-Myc, c-Jun and Usp28 following FT206 treatment.

We found that the ubiquitination levels of c-Myc and c-Jun increased upon FT206 treatment, confirming that FT206 blocks USP28-mediated deubiquitination of its substrates (new Figure S4B). We also found that the ubiquitination level of USP28 increased upon FT206 treatment (new Figure S4B), which is consistent with previous observations where the enzymatic activity of DUBs is required to enhance their own stability (https://doi.org/10.1038/cdd.2011.16).

Reviewer #3:The prevalent treatment options for LSCC are limited in efficacy. Through genetic inactivation of Usp28 in a novel lung cancer mouse model, and chemical inhibition of Usp28 in induced LSCC in mice and human LSCC xenograft tumors, the authors demonstrated the specific dependency of LSCC (but not LADC) on the protein deubiquitinase Usp28. The authors also showed that loss of Usp28 by either means leads to depletion of the oncoproteins c-Myc, p63 and c-Jun in LSCC. Finally, the authors described a novel small molecule that is specific for Usp25/28. Based on these results, the authors suggested chemically targeting USP28 as a potential therapeutic option for human LSCC patients.Strengths: The presentation of the work is clear, concise and easily readable. The data presented largely supports the authors' conclusions on the role of USP28 in LSCC tumorigenesis and that inhibition of USP28 is a viable therapeutic option for LSCC treatment. The generation of the KFCU mice model that can give rise to both LADC and LSCC concurrently is interesting and presents a valuable tool for the wider cancer community.Weakness: The manuscript can benefit from a deeper analysis of the relationship between FBW7 and USP28 in patient cohorts. A comparison of the activity/efficacy of FT206 to existing USP28 inhibitors will also be helpful.1. The authors mentioned that 25% of human lung squamous cell carcinoma cases show gain of function alterations in USP28, based on TCGA data.– What is the proportion of cases in the current study cohort (n=17) which show similar gain of function alterations at the DNA level, as well as overexpression of USP28 at the protein level (by immunohistochemistry or immunoblotting)?

We would like to thank also this reviewer for insightful comments. Unfortunately, we cannot evaluate DNA genetic alterations in our study cohort as we do not have any additional remaining sample left to extract DNA. However, as requested, we have performed immunohistochemistry (IHC) analysis for USP28, and found that LSCC tumours express very high levels of USP28 protein (new Figure S2B).

However, we have mRNA from these analysed samples, and we found that in our study cohort low USP28 mRNA levels correlate with low Usp28 protein levels and likewise, high/moderate mRNA levels also correlate with high Usp28 protein levels (Figure 1G and new Figure S2B).

-What is the correlation between DNA alterations and mRNA/protein expression? This would be clinically relevant if USP28 inhibitors are to be used in clinic, since we need a robust predictive test to select for patients who are most likely to respond to this therapy.

This is an excellent suggestion. We have performed this analysis using TCGA data, which revealed that there is a positive correlation between *USP28* copy-number gain and higher mRNA expression in human LSCC patients (new Figure S2A). We completely agree with the insightful suggestion by this referee using USP28 status for potential LSCC patient selection for USP28 treatment. It will be very interesting to determine if USP28 inhibitors will have higher efficacy in patients with gain-of-function alterations in USP28.

2. A previous study by the same first author mentioned that 69% of patients with LSCC show loss of FBW7 expression by immunohistochemistry.– What is the FBW7 status of the cohort in the current study?

As requested, we have performed immunohistochemistry analysis for FBW7, and found that 16/17 (94%) LSCC tumors express low levels of FBW7protein (new Figure S2B), which is even higher than in the cohort we previously reported.

– Can USP28 overexpression/gain of function co-exists with FBW7 loss, or are they mutually exclusive?

Analysing LSCC TCGA data, we found that 44/178 human LSCC cases show overexpression/gain of function in USP28 (~25%). Interestingly, 30 of those 44 cases (68.1%) also display loss of FBXW7. Thus, in a significant fraction of LSCC patients, USP28 overexpression/gain of function co-exists with FBXW7 loss*.* However, ~75% of cases with FBXW7 alterations do not show USP28 gain-of-function (new Figure 1F). Thus, USP28 overexpression/gain-of-function and FBXW7 loss are not mutually exclusive.

Moreover, in our cohort we found by IHC that all human LSCC samples express USP28 (mostly very strongly; new Figure S2B) concomitant with FBW7 loss (new Figure S2B). Thus, USP28 overexpression co-exists with FBW7 loss in human LSCC patients.

– Apart from USP28 gain of function, FBW7 loss of function may also predict sensitivity to USP28 inhibition. Related to this, what is the status of FBW7 and USP28 in the three human LSCC cell lines used in the xenograft studies? This would clarify if the observed effect for USP28 inhibition is specific to LSCC cell lines with USP28 overexpression/FBW7 loss or to LSCC cell lines in general, regardless of USP28/FBW7 status.

We have analysed COSMIC (https://cancer.sanger.ac.uk/cosmic) and canSAR

(https://cansarblack.icr.ac.uk/) genetic databases and found that all 3 human LSCC cell lines (NCI-H520, CALU-1 and LUDLU-1) used in our study do not show mutations in FBXW7 nor USP28. An example is shown in Author response image 1 (NCI-H520, (Author response image 1); we only added this Figure to the rebuttal letter, not the manuscript, as this information is publicly available, but we would include it in the manuscript if this referee felt this was useful). Thus, these data support the notion that LSCC cells respond to USP28 inhibition, regardless of USP28/FBXW7 mutation status, which suggest that USP28 inhibition might be a therapeutic option for all LSCC patients.

**Author response image 1. sa2fig1:** Analysis in COSMIC (panel A) or canSAR (panel B) databases showing no evidence of mutations in USP28 nor FBXW7 genes.

3. It is interesting that LADC is not affected by the loss of Usp28. What is c-MYC and c-Jun protein expression in the LADC lesions in the KFCU mice? Are they upregulated upon KrasG12D activation and Fbw7 deletion? Although the authors showed in Figure 1 that the expression of c-MYC is lower in LADC compared to LSCC, it will be important to directly assess whether loss of USP28 (either by siRNA knockdown or FT206 treatment) in LADC cell lines can affect c-MYC protein expression.

Our Western blot analysis revealed that Usp28 deletion resulted in reduced c-Jun and c-Myc protein levels in KFCU LADC lesions, although the reduction in c-Myc protein levels appear to be significantly less pronounced than observed in LSCC (new Figure 2E).

The modest decrease in Usp28 substrates could explain the modest reduction in LADC tumor size.

In addition, human LADC cell lines were treated with FT206 (IC50 doses) and c-Myc/c-Jun levels determined. We found that FT206 treatment reduced c-Myc (>65%) and c-Jun (>80%) protein levels in the three human LSCC cell lines used in the xenograft studies (new Figure S6A).

Although Usp28 inhibition also decreased c-Myc levels in 2/3 LADC lines, the reduction in cMyc protein levels appear to be significantly less pronounced than observed in LSCC (new Figure S6B). In contrast, treatment of LADC cells with FT206 resulted in a significant increase in c-Jun protein levels. Thus, Usp28 inhibition acts differently in LADC and LSCC tumor cells.

4. As an important part of the paper is about application of the new Usp28 inhibitor FT206, the authors should have compared the efficacy with previously described Usp25/28 inhibitor (Wrigley et al; 2017, ACS Chem Biol 12,3113-3125) or at the very least, comment on potential similarities/differences/improvements etc.

We completely agree with this referee that this will be a useful addition to our study. To this end, we have compared USP28 cellular target engagement properties of the AZ1 and FT206 compounds using our activity-based protein profiling assay with a Ub-based probe as described for Figure 4B, and have added this data in new Figure S4A.

5. As the authors advocate targeting USP28 in LSCC treatment, have they tested, or can they comment on, whether USP28 inhibition would be beneficial for LSCC that do not have gain-of-function alterations in USP28 (which represent 75% of LSCC)?

The three human LSCC cell lines (NCI-H520, CALU-1 and LUDLU-1) used in the xenograft experiment in our study, each of which that responded well to USP28 inhibition, do not show gain-of-function mutations in USP28 (Author response image 1). Thus, these data suggest that targeting USP28 could be beneficial for LSCC patients that do not have gain-of-function alterations in USP28.

[Editors’ note: what follows is the authors’ response to the second round of review.]

Reviewer #2:Overall the authors have addressed most of my concerns in the previous submission. I find the revised manuscript much clearer and significantly improved. I only have 4 additional points for the authors' clarification:1) In light of the new data showing that FT206 can also inhibit USP25, albeit to a lesser extent than USP28, I think it is important to address if depletion of USP25 also results in loss of c-Myc, c-Jun and ∆p63 expressions compared to USP28 knockdown, at least in cell lines.

This is an important point. As suggested, we knockdown the expression of Usp25 by siRNAs and observed that Usp25 downregulation did not reduced protein levels of c-Jun, p63 and cMyc in LSCC cells (new Figure 6 —figure supplement 1A). Thus, together this data suggests that the observed effects of FT206 treatment are mainly mediated by Usp28.

Related to this, what is the phenotype of USP25 KO mice (if known)?

USP25 KO mice are viable and do not show any abnormalities in growth or survival (Nat Immunol 2012, 13:1110) but are more susceptible to H5N1 or HSV-1 viral infection compared to their wild-type counterparts (PNAS 2015,112:11324). This is linked to USP25 association with TRAF3 and TRAF6, main components of innate immune response, and no apparent functional connections appear to exist with c-Myc, c-Jun or p63 (PNAS 2015,112:11324).

2) The authors mentioned that inhibitor treated mice kept a normal body weight, indicating no global adverse effects (Figure S5A). The numbers of mice used (n = 3) are on the low side. Do the authors have body weight and survival information for the other FT206 treated genetic mouse models (e.g Figure 2)?

As requested, we now show the body weight for the genetic models displayed in Figure 2 (new Figure 2 —figure supplement 1E, n = 7-8). We observed a transient loss of body weight during Tamoxifen treatment, yet body weight recovered a few days later. This effect has been found in other studies that use Tamoxifen to delete floxed alleles (Cancer Cell 2019; 35:573), as it is well-known that Tamoxifen transiently decreases food intake, body weight and in the short-term fat mass in rodents (Am J Physiol 1993 264:R1219). Importantly, FT206 administration did not result in a further reduction of body weight (new Figure 2 —figure supplement 1E).

Unfortunately, we do not have survival curves for these experiments. Mice from different groups (i.e. Vehicle, Tam and Tam+FT) were culled at the same time to quantify side-by-side the number of lung tumours. However, the decreased tumour burden seen in lung tumour in the absence of Usp28 suggested that survival will be impacted. To support the clinical significance of targeting Usp28, we examined the correlation between USP28 expression level and LSCC patient survival (Author response image 2). Lower expression of USP28 is associated with a significantly longer survival time (*P* = 8.4x10^-3^).

**Author response image 2. sa2fig2:** Kaplan-Meier plot showing the association between *USP28* expression and patient survival. Analysis performed using KM plotter lung cancer database.

3) The authors showed in Figure S2A that there is a positive correlation between USP28 copy number gain and higher mRNA expression in human LSCC patients, which may be useful for patient selection. However, they also mentioned that the 3 LSCC cell lines used for the xenograft studies do not show gain-of-function mutations in USP28 but responded well to USP28 inhibition. Do the authors have any data comparing the effects of USP28 deletion and/or inhibition in LSCC cell lines with or without USP28 alterations?

This is an interesting point. We believe that both situations are not mutually exclusive. Whereas those patients with gain-of-function alterations in USP28 might have a higher response to USP28 inhibitors, our xenograft data suggest that patients without USP28 alterations could also benefit from USP28 inhibition.

As requested, we compared the effects of USP28 inhibition in LSCC cell lines with or without USP28 alterations. In Figure 6 —figure supplement 1B we found that FT206 treatment resulted in a significant decrease of c-Myc and c-Jun protein levels in three human LSCC cell lines (H520, CALU-1 and LUDLU-1) that do not have USP28 genetic alterations. The LSCC cell line SKMES contains a Nonsense Mutation in USP28 (c.193G>T), resulting in a nonfunctional protein product. Consequently, FT206 treatment failed to decrease c-Myc and cJun protein levels in this cell line (new Figure 6 —figure supplement 1E). This data further confirms that the effects of FT206 are mediated by USP28 and also suggests that patients with loss-of-function alterations in USP28 will not respond to USP28 inhibitors.

4) Related to point 3, can the authors speculate or discuss why USP28 deletion/inactivation has a less pronounced effect in LADC despite the latter also having high expression of USP28 and c-MYC ?

Our Western blot analysis revealed that Usp28 deletion resulted in reduced c-Myc and c-Jun protein levels in LADC lesions, but, importantly, the reduction in c-Myc protein levels appears to be significantly more pronounced in LSCC (Figure 2E). Moreover, Usp28 ablation has a marginal effect, if any, on apoptotic cell death (cleaved caspase-3; CC3). Thus, the modest decrease in Usp28 substrates and the failure to induce cleaved caspase-3 could explain the modest effect in LADC tumours. We have added a paragraph in the Results section that mentions these characteristics (page 8, lines 2-6).